# A size principle for recruitment of *Drosophila* leg motor neurons

**Anthony W Azevedo[1], Evyn S Dickinson[1], Pralaksha Gurung[1], Lalanti Venkatasubramanian[2], Richard S Mann[2], John C Tuthill[1]***

[1]Department of Physiology and Biophysics, University of Washington, Seattle, United States; [2]Department of Biochemistry and Molecular Biophysics, Department of Neuroscience, Zuckerman Mind Brain Behavior Institute, Columbia University, New York, United States

**Abstract** To move the body, the brain must precisely coordinate patterns of activity among diverse populations of motor neurons. Here, we use in vivo calcium imaging, electrophysiology, and behavior to understand how genetically-identified motor neurons control flexion of the fruit fly tibia. We find that leg motor neurons exhibit a coordinated gradient of anatomical, physiological, and functional properties. Large, fast motor neurons control high force, ballistic movements while small, slow motor neurons control low force, postural movements. Intermediate neurons fall between these two extremes. This hierarchical organization resembles the size principle, first proposed as a mechanism for establishing recruitment order among vertebrate motor neurons. Recordings in behaving flies confirmed that motor neurons are typically recruited in order from slow to fast. However, we also find that fast, intermediate, and slow motor neurons receive distinct proprioceptive feedback signals, suggesting that the size principle is not the only mechanism that dictates motor neuron recruitment. Overall, this work reveals the functional organization of the fly leg motor system and establishes *Drosophila* as a tractable system for investigating neural mechanisms of limb motor control.

*For correspondence: tuthill@uw.edu

Competing interests: The authors declare that no competing interests exist.

## Introduction

Dexterous motor behaviors require precise neural control of muscle contraction to coordinate force production and timing across dozens to hundreds of muscles. This coordination is mediated by populations of motor neurons, which translate commands from the central nervous system into dynamic patterns of muscle contraction. Although motor neurons are the final common output of the brain, the scale and complexity of many motor systems have made it challenging to understand how motor neuron populations collectively control muscles and thus generate behavior. For example, a human leg is innervated by over 150,000 motor neurons and a single calf muscle is innervated by over 600 motor neurons (*Kernell, 2006*). How can the nervous system coordinate the activity of such large motor neuron populations to flexibly control the force, speed and precision of limb movements?

One way to streamline motor control is to establish a hierarchy among neurons controlling a particular movement, such as flexion of a joint. This hierarchy allows premotor circuits to excite different numbers of motor neurons depending on the required force: motor neurons controlling slow or weak movements are recruited first, followed by motor neurons that control progressively stronger, faster movements (*Denny-Brown, 1929*). A recruitment order for vertebrate motor neurons innervating a single muscle was first postulated over 60 years ago (*Henneman, 1957*). Subsequent work identified mechanisms associated with the recruitment order and synthesized these findings as the *size principle*, which states that small motor neurons, with lower spike thresholds, are recruited prior to larger neurons, which have higher spike thresholds (*Henneman et al., 1965a*; *Henneman et al., 1965b*; *Henneman and Olson, 1965*; *Mcphedran et al., 1965*; *Mendell, 2005*; *McPhedran et al.,*

**eLife digest** In the body, spindly nerve cells called motor neurons connect the brain to the muscles. Their role is to control movement, as they translate the electrical signals from the brain into instructions to the muscles. In humans, it takes over 150,000 motor neurons to control the movement of one leg; in contrast, fruit flies only need 50 neurons to operate a leg, despite also executing a variety of movements.

Fruit flies are commonly used in laboratories to study an array of biological processes, yet little is known about how their motor neurons direct movements. In particular, it was unclear whether the same principles that control how muscles contract in mammals also applied in the tiny fruit fly.

To begin investigating, Azevedo et al. mapped out the arrangement of motor neurons that control muscles in the fruit fly leg. As the leg moved, the activity of both the neurons and the muscles they controlled was recorded, as well as the force that had been generated.

The experiments showed that each motor neuron controls a certain range of leg force and speed: some produced small, slow motion important for posture and dexterity, while others created large, fast movements essential to running or escape. In addition, the neurons activate in a particular order – cells that control slow movements fire first, and those that direct fast maneuvers later. These processes are also found in other organisms, but the difference is that flies have so few neurons, allowing scientists to reliably identify each motor neuron. Future experiments will therefore be able to test how flies recruit the right neurons to create specific movement sequences.

Fruit flies are often used to research human illnesses that affect movement, such as motor neuron disease. A better understanding of the way their neural circuits coordinate the body could help reveal how these conditions emerge.

*1965*). Evidence for the size principle has been provided by electrophysiological analysis of motor neurons in a number of species, from crayfish to humans (*Gabriel et al., 2003*; *Hill and Cattaert, 2008*; *McLean and Dougherty, 2015*; *Milner-Brown et al., 1973*; *Sasaki and Burrows, 1998*). These studies have also described systematic relationships between motor neuron electrical excitability, recruitment order, conduction velocity, force production, and strength of sensory feedback and descending input (*Bawa et al., 1984*; *Binder et al., 1983*; *Fleshman et al., 1981*; *Heckman and Binder, 1988*; *Kernell and Sjöholm, 1975*; *Zengel et al., 1985*).

A simplifying assumption of the size principle is that all motor neurons within a pool receive identical presynaptic input, such that recruitment order is entirely dictated by the gradient of motor neuron excitability. However, recordings in vertebrates have found that presynaptic inputs may vary within a motor pool (*McLean and Dougherty, 2015*) and that recruitment order can change during specific movements (*Desnedt and Gidaux, 1981*; *Kishore et al., 2014*; *Menelaou and McLean, 2012*; *Smith et al., 1980*). Are these violations of the size principle a fundamental feature of motor control circuits or rare exceptions to the rule? Answering this question would be greatly aided by a tractable system in which it was possible to genetically target identified motor neurons for recording, manipulation, and mapping of presynaptic inputs.

The leg of the fruit fly, *Drosophila melanogaster*, contains 14 muscles which are innervated by just 53 motor neurons (*Baek and Mann, 2009*; *Brierley et al., 2012*; *Maniates-Selvin et al., 2020*; *Soler et al., 2004*). In spite of this tiny scale, the fly leg supports a variety of fast and flexible behaviors. During forward walking, the fly grips the substrate with the distal segment of its front leg (the tarsus) and flexes the femur-tibia joint, pulling the body forward (Video 4). The femur-tibia joint of a walking fly flexes and extends 10–20 times per second, reaching swing speeds of several thousand degrees per second (*DeAngelis et al., 2019*; *Gowda et al., 2018*; *Mendes et al., 2013*; *Strauss and Heisenberg, 1990*; *Wosnitza et al., 2013*). Flies also use their legs to target other body parts during grooming (*Hampel et al., 2015*; *Seeds et al., 2014*), for social behaviors like aggression and courtship (*Clowney et al., 2015*; *Hoopfer et al., 2015*), and to initiate flight take-off (*Card and Dickinson, 2008*; *Zumstein et al., 2004*) and landing (*Ache et al., 2019*). These behaviors require a wide range of muscle force production across multiple timescales. However, little is currently known about the organization and function of leg motor control circuits in the fly. While a great deal of progress has been made on understanding the processing of sensory signals in the

*Drosophila* brain, we lack an understanding of how this information is translated into behavior by the fly's ventral nerve cord (VNC). Investigating motor control in *Drosophila* is important because the fly's compact nervous system and identified cell types make it a tractable system for comprehensive circuit analysis (*Tuthill and Wilson, 2016a*).

In this study, we investigate motor control of the *Drosophila* tibia. We first mapped the organization of tibia flexor motor units using calcium imaging from leg muscles in behaving flies (Figure 1). With electrophysiology, we then discovered that motor neurons controlling tibia flexion (Figure 2) lie along a gradient of anatomical and physiological properties (Figure 3) that correlate with muscle force production (Figure 4). *Slow* motor neurons produce <0.1 µN per spike, while *fast* motor neurons produce ~10 µN per spike, approximately equal to the fly's weight. Recordings during spontaneous leg movements revealed a recruitment hierarchy: slow motor neurons typically fire first, followed by intermediate, then fast neurons (Figure 5). Interestingly, all tibia flexor motor neurons receive feedback from proprioceptors at the femur/tibia joint, but these sensory signals vary in amplitude, sign, and dynamics across the different motor neuron types (Figure 6). Optogenetic manipulation of each motor neuron type had unique and specific effects on the behavior of walking flies (Figure 7), consistent with their roles in controlling distinct force regimes.

Together, these data establish the organization and function of a key motor control module for the fly leg. Overall, we found that motor neurons controlling the fly tibia exhibit many features consistent with the size principle. However, we also observed that tibia motor neurons receive distinct proprioceptive feedback signals and that recruitment order is occasionally violated. Thus, in addition to the size principle, heterogeneous input from premotor circuits is likely to play an important role in coordinating neural activity within a motor pool.

## Results

### Organization and recruitment of motor units controlling the femur-tibia joint of *Drosophila*

We first sought to understand how muscles in the fly femur control movement of the tibia (*Figure 1*). Fly leg muscles are each composed of multiple fibers (*Soler et al., 2004*) and innervated by distinct motor neurons (*Baek and Mann, 2009*; *Brierley et al., 2012*). A motor neuron and the muscle fibers it innervates are referred to as a *motor unit*. In most invertebrate species, multiple motor neurons can innervate the same muscle fiber, so motor units may be overlapping (*Hoyle, 1983*).

We used widefield fluorescence imaging of muscle calcium signals with GCamp6f (*Chen et al., 2013*; *Lindsay et al., 2017*) to map the spatial organization of motor units in the fly femur during spontaneous tibia movements (*Figure 1C*, *Figure 1—figure supplement 1*, and *Video 1*). We observed that spatial patterns of calcium activity were different for different movements (e.g., tibia flexion vs. extension).

To quantify this spatial organization, we performed unsupervised clustering so that pixels with correlated activity were grouped together into *activity clusters*. We assigned arbitrary numbers to these clusters, from ventral to dorsal (*Figure 1—figure supplement 1*). Three clusters significantly increased their fluorescence when the leg was flexed (*Figure 1C*); we refer to these clusters as Flexors 1, 2, and 3. Flexors 2 and 3 were active during most periods of tibia flexion, whereas Flexor 1 was only active only during large, fast movements (*Figure 1E*). This organization was similar across flies and robust to changes in clustering parameters (*Figure 1—figure supplement 1*). Likely because of the orientation of the leg in these experiments, we did not observe any clusters whose activity increased specifically during tibia extension (see *Figure 1—figure supplement 1* and Materials and Methods for further discussion). For the remainder of this study, we focus our efforts on motor control of tibia flexion.

For Flexors 1 and 2, we could record and identify extracellular motor neuron spikes in the femur (electromyogram: EMG), which confirmed that the activity clusters identified through calcium imaging correspond to distinct motor units (*Figure 1E* and see *Figure 2D*, below). In other words, we propose that activity clusters 1 and 2 reflect the firing patterns of specific motor neurons. Below, we elaborate on this relationship further when discussing specific motor neurons.

Muscles controlling the same joint may exert different levels of force, depending on fiber type composition and musculoskeletal biomechanics. To understand the relationship between motor unit

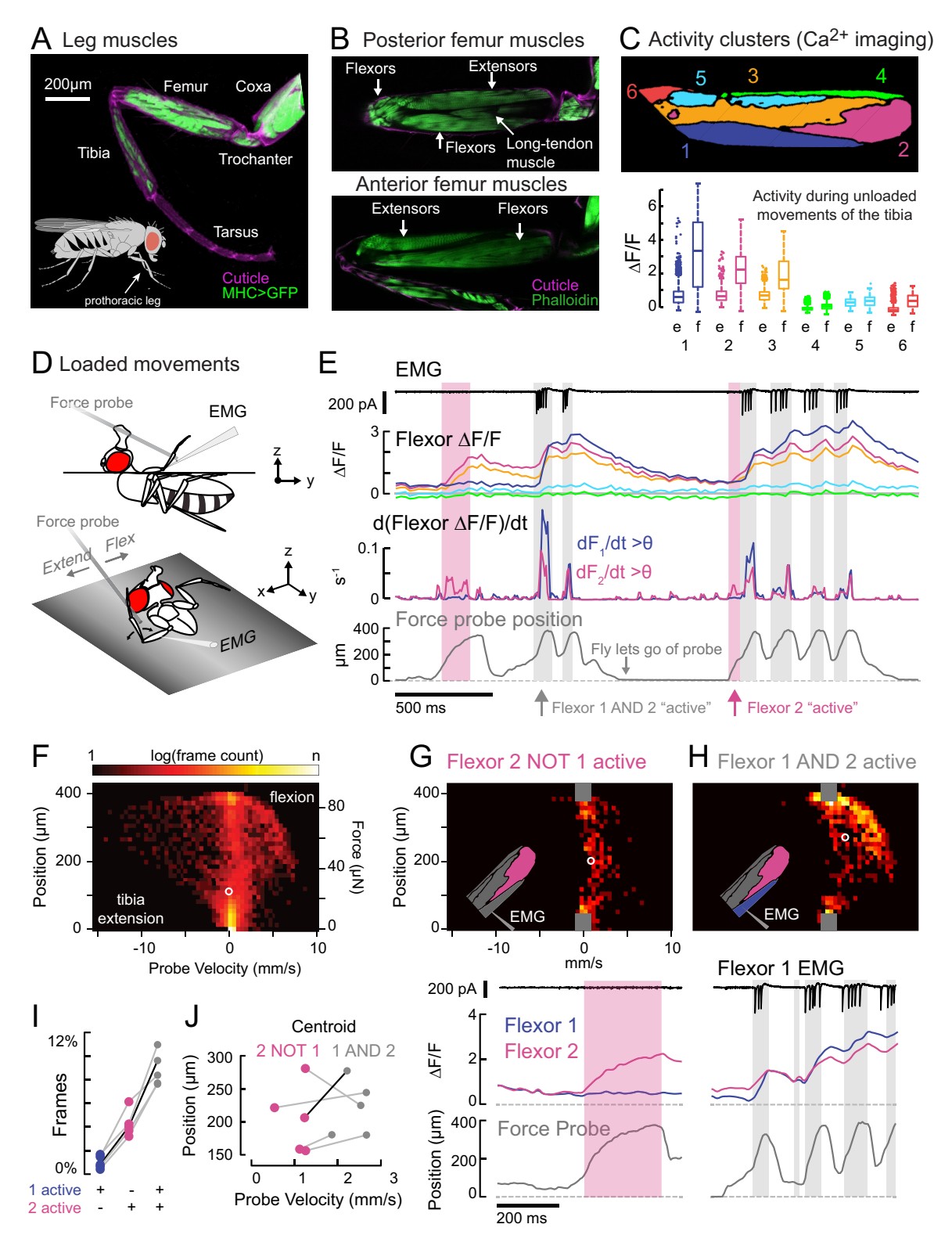

**Figure 1.** Functional organization and recruitment order among muscles controlling the *Drosophila* leg. (A) Muscles of the right prothoracic leg of a female *Drosophila* (*MHC-LexA; 20XLexAop-GFP*). (B) Muscles controlling tibia movement in the fly femur. Top and bottom are confocal sections through the femur. Anterior-posterior axis refers to the leg in a standing posture (***Soler et al., 2004***). (C) Top: K-means clustering of calcium signals (*MHC-Gal4;UAS-GCaMP6f*) based on correlation of pixel intensities during 180 s of self-generated leg movements in an example fly (unloaded waving, *Figure 1 continued on next page*

*Figure 1 continued*

see methods). Bottom: average change in fluorescence for each cluster, in each frame, when the leg is extended (femur-tibia joint >120°, 20772 frames) vs. flexed (<30°, 61,860 frames), n = 5 flies. Flexion activity was consistently higher (p=0.01 for cluster 5, p<10$^{-6}$ for all other clusters, 2-way ANOVA, Tukey-Kramer correction). (D) Schematic of the experimental setup. The fly is fixed in a holder so that it can pull on a calibrated force probe with the tibia while calcium signals are recorded from muscles in the femur. (E) Calcium activity in tibia muscles while the fly pulls on the force probe (bottom trace). Cluster six was obscured by the probe and not included. The middle row shows the smoothed, rectified derivative of the cluster fluorescence (dF$_i$/dt) for the two brightest clusters (1 and 2), which we refer to as Flexors. Highlighted periods indicate that both Flexors 1 and 2 are active simultaneously (gray, dF/dt > 0.005), or that Flexor 2 alone is active (magenta). (F) 2D histogram of probe position and velocity, for all frames (n = 13,504) for a representative fly. The probe was often stationary (velocity = 0), either because the fly let go of the probe (F = 0), or because the fly pulled the probe as far as it could (F ~ 85 µN), reflected by the hotspots in the 2D histogram. In F–H), the white circles indicate the centroids of the distributions. (G) Top - 2D histogram of probe position vs. velocity when Flexor 2 fluorescence increased, but not Flexor 1 (n = 637 frames, same fly as F). Gray squares indicate hotspots in F), which are excluded here. Color scaled to log(50 frames). Bottom – example of instance in which Flexor 2 alone is active (magenta shading). (H) Top - Same as G), when Flexor 1 AND 2 fluorescence increased simultaneously (n = 1449 frames). Bottom – gray shading indicates instances of activity in both Flexors 1 and 2. (I) Fraction of total frames for each fly in which both Flexor 1 and 2 fluorescence increased (gray), Flexor 2 alone increased (magenta), or Flexor 1 alone increased. Number of frames for each of five flies: 32,916, 13,504, 37,136, 37,136, 24,476. (J) Shift in the centroid of the 2D histogram when Flexor 1 fluorescence is increasing along with Flexor 2 fluorescence (gray), compared to when Flexor 2 fluorescence alone is increasing (p<0.01, Wilcoxon rank sum test). Black line indicates example cell in G) and H).

The online version of this article includes the following figure supplement(s) for figure 1:

**Figure supplement 1.** Wide-field calcium imaging of muscles in the femur.
**Figure supplement 2.** Calibration of the force probe dynamic properties.
**Figure supplement 3.** Flexor 2 signal is high whenever Flexor 1 is active.
**Figure supplement 4.** GFP control for k-means clustering of calcium activity.

activity and force production, we allowed the fly to pull on a flexible probe with its tibia (*Figure 1D*). We calibrated the force required to deflect the probe a given distance and measured a spring constant of 0.22 µN/µm (*Figure 1—figure supplement 2*). Flies were able to move the force probe up to 400 µm, reaching speeds of 8 mm/s (*Figure 1F*). That is, the fly was capable of producing close to 100 µN of force at the tip of the tibia and changes in force of ~1.3 mN/s. For comparison, the mass of the fly is ~1 mg for a weight of ~10 µN; this means that the femur-tibia joint can produce enough force to lift approximately ten times the fly's body weight.

We used the activity clusters computed from unloaded leg movements (*Figure 1C*) to examine whether different flexor motor units control different levels of tibia force production. We observed that Flexor 2 activity increased across a range of tibia velocities and forces, but Flexor 1 activity increased only during the fastest, most powerful movements. To quantify this relationship, we compared probe force and velocity when clusters were recruited either alone or together (*Figure 1G–J*). The kinetics of GCaMP6f are slow relative to fly tibia movements, so we examined only periods when the derivative of the fluorescence signal was positive, *i.e.* muscle contraction was increasing (*Figure 1E*). The highest probe forces and velocities were achieved only when both Flexors 1 and 2 were active together (*Figure 1G–H*). When Flexor 2 was active alone, probe velocities were always lower (*Figure 1J*). Occasionally, the derivative of Flexor 1 fluorescence alone was high (*Figure 1I*), but in these rare instances the intensity of Flexor 2 was also high (*Figure 1—figure supplement 3*), indicating that Flexor 2 was contracting. These results indicate that

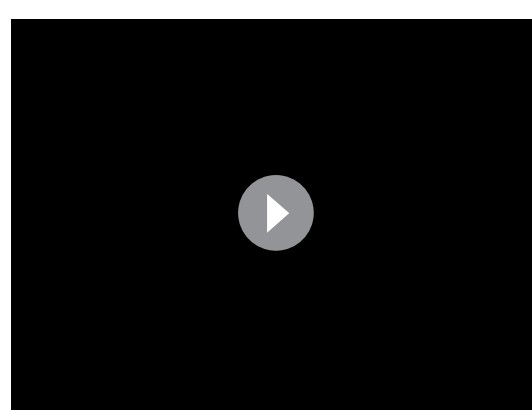

**Video 1.** Calcium imaging of femur muscles. The video (1) illustrates the arrangement of musculature controlling the fly's tibia; (2) schematizes the position of a restrained fly relative to the force probe fiber, together with wide-field calcium imaging from muscles in the leg and body; and then (3) shows simultaneous videos of the fly pulling on the force with muscle calcium signals in the femur, together with the movement of the probe, an EMG recording from the fast neuron, and the time course of cluster ΔF/F. Videos are slowed 3X.

https://elifesciences.org/articles/56754#video1

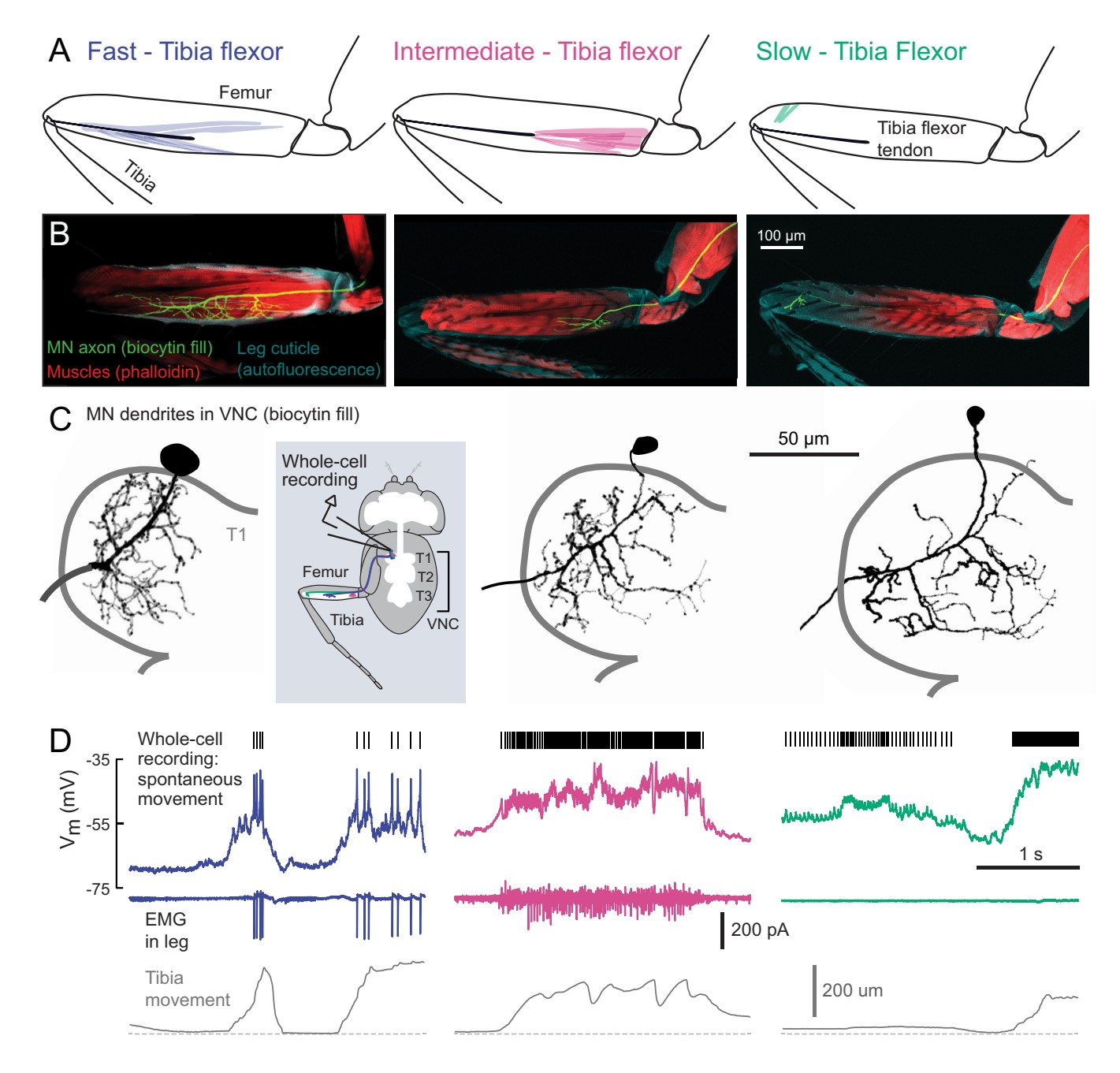

**Figure 2.** Motor neurons controlling tibia flexion. (A) Schematic of the muscle fibers innervated by motor neurons labeled by the following Gal4 lines: left: *R81A07-Gal4*, center: *R22A08-Gal4*, and right: *R35C09-Gal4*. (B) Biocytin fills of leg motor neurons from whole-cell patch clamp recordings, labeled in fixed tissue with strepavidin-Alexa-568 (green), counter stained with phalloidin-Alexa-633 (red) labeling actin in muscles, and autofluorescence of the cuticle (cyan). (C) Images of biocytin fills in the VNC following whole-cell patch recordings (schematic). Prothoracic neuromere is outlined. Cells were traced and 'filled' with the simple neurite tracer plugin in Fiji. (D) Example recordings from each motor neuron type during spontaneous tibia movement: whole-cell current clamp recordings with detected spike times above (top row, corrected for a liquid junction potential of −13 mV), EMG recordings (middle), and tibia movement (force probe position; bottom). We often observed EMG events from the fast and intermediate neurons. We did not observe EMG events associated with slow neuron firing, though we often observed flexion of the tibia without seeing EMG activity, as seen here.

The online version of this article includes the following figure supplement(s) for figure 2:

**Figure supplement 1.** Central anatomy of flexor motor neurons.

distinct motor units control distinct levels of force production, and that they are recruited in a specific order, with motor units controlling low forces firing prior to motor units controlling higher forces.

The results of *Figure 1* illustrate two organizational features of fly leg motor control. First, fly tibia flexion is controlled by a number of distinct motor units in the femur that are active at different levels of force production. Second, the sequential activity of tibia flexor motor units is consistent with a hierarchical recruitment order. The spatial organization of tibia flexor motor units also provides a template that can be used to identify genetic driver lines that label specific motor neurons.

## A gradient of electrophysiological and anatomical properties among motor neurons controlling flexion of the femur-tibia joint

We visually screened a large collection of Gal4 driver lines (*Jenett et al., 2012*) for expression in motor neurons that innervate the activity clusters we identified through calcium imaging (*Figure 1*). We focus our analysis on three lines that each label a single tibia flexor motor neuron in the femur. The first line (*R81A07-Gal4*) labels a motor neuron that innervates the high-force motor unit (*Figure 2*, left) that we identified through calcium imaging (Flexor 1). The second (*R22A08-Gal4*) labels a neuron that projects to the proximal femur (*Figure 2*, middle), one of several likely candidates that controls Flexor 2 (*Baek and Mann, 2009*). The muscle fibers innervated by these two neurons connect to the same tibia flexor tendon (*Soler et al., 2004*). The third line (*R35C09-Gal4*) labels a single motor neuron that projects to the distal part of the femur, where the signals in our calcium imaging experiments were weak and noisy (*Figure 2*, right). The muscle fibers in this distal region have been collectively referred to as the *reductor* muscle (*Baek and Mann, 2009*; *Brierley et al., 2012*; *Soler et al., 2004*), but their alignment and attachment points suggest they control flexion of the tibia (see Materials and Methods for further details).

For consistency with the literature on other insects (*Burrows, 1996*), and based on their functional properties described below, we refer to these three motor neurons as *fast*, *intermediate*, and *slow*. Previous studies of motor neuron development suggest that the fast motor neuron is a unique, embryonically born neuron that persists through metamorphosis, whereas the intermediate and slow motor neurons belong to larger cohorts born during metamorphosis (*Baek and Mann, 2009*; *Brierley et al., 2012*). Those studies estimated that 2–5 neurons innervate the proximal muscle fibers in the femur, similar to the intermediate motor neuron, and 8–9 neurons innervate distal muscle fibers, similar to the slow motor neuron.

Using in vivo whole-cell patch-clamp electrophysiology, we recorded the membrane potential of each motor neuron while the fly pulled on the force probe. The recording configuration was similar to that used for calcium imaging (*Figure 1*). We observed increases in the firing rate of each motor neuron type during tibia flexion (*Figure 2D*). Simultaneous EMG recordings from motor neuron axons in the femur confirmed that the fast motor neuron corresponds to Flexor 1, identified via calcium imaging (*Figure 1*). The intermediate neuron contributes to the activity of Flexor 2, likely along with one to two other similar intermediate neurons. Perhaps because of the small size of its axon, we were unable to identify slow motor neuron spikes with extracellular recordings.

The three identified motor neurons have conspicuous differences in the size of their axons, dendrites, and cell bodies (*Figure 3A*). The fast motor neuron has an exceptionally large soma (for a *Drosophila* neuron) and thick dendritic branches. Its axon has a wide diameter and branches extensively in the femur (*Figure 2B*). The intermediate motor neuron has a smaller cell body, thinner dendritic branches, and smaller axonal arborization in the femur. The slow motor neuron has the smallest cell body, dendrites, and axon of the three. Each motor neuron has a unique dendritic branching pattern within the VNC neuropil. For example, the intermediate neuron projects toward the midline, and the arborizations of the fast and intermediate neurons branch more extensively than the slow neuron (*Figure 2C*, *Figure 2—figure supplement 1*). Thus, each motor neuron is anatomically positioned to receive distinct presynaptic inputs.

The intrinsic electrical properties of the tibia flexor motor neurons varied along a continuum (*Figure 3B–E*). The average resting potential of fast motor neurons was lower (−68 mV) than that of intermediate (−60 mV) and slow (−48 mV) motor neurons (*Figure 3C*). While the fast and intermediate neurons were silent at rest, the slow neuron had a resting spike rate of approximately 30 Hz (*Figure 3D*). We also observed a gradient in input resistance, which we calculated from the voltage responses to small hyperpolarizing current injections before each trial (−5 pA). The input resistance

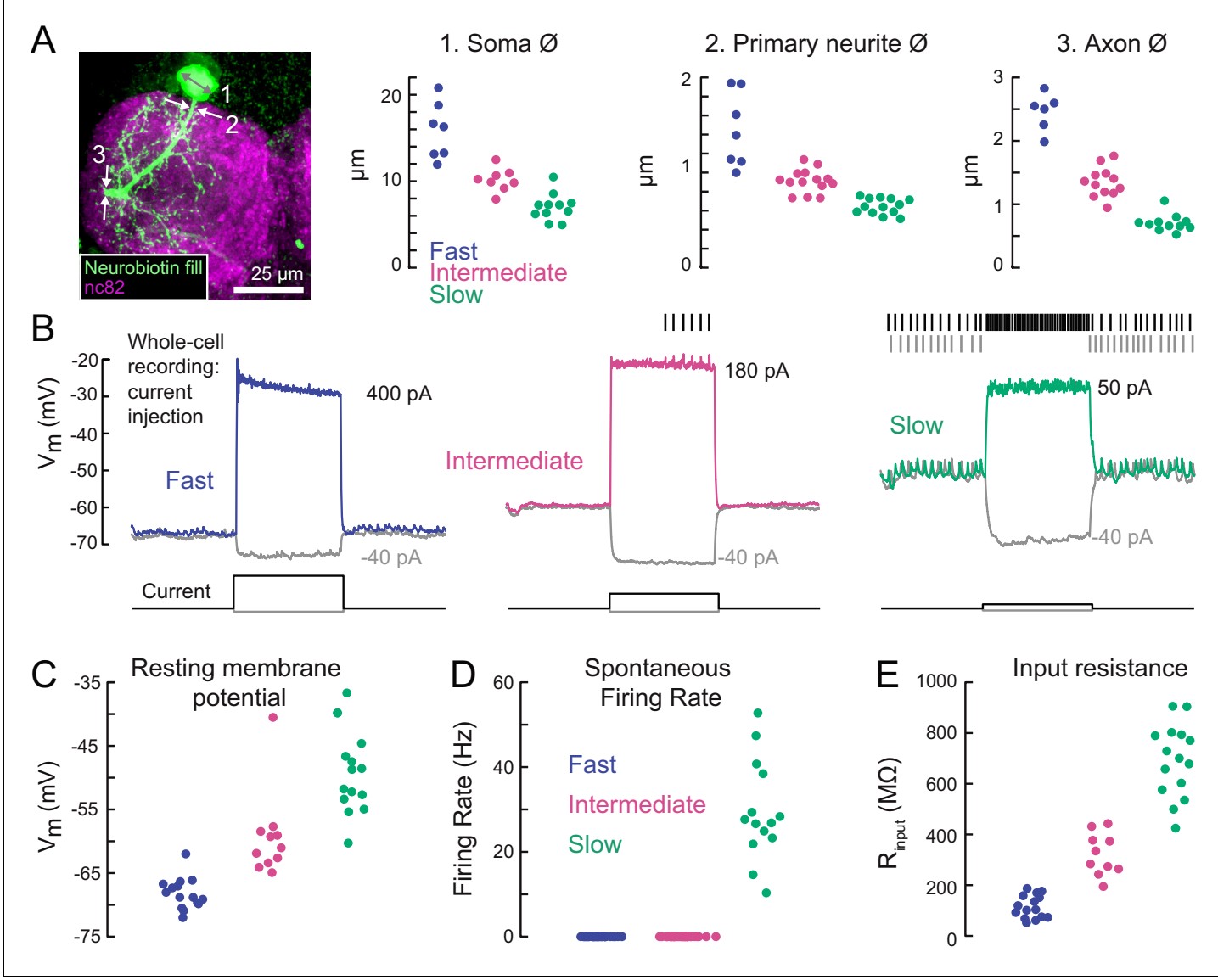

**Figure 3.** A gradient of intrinsic properties among tibia flexor motor neurons. (**A** Left) Maximum intensity projection of a Neurobiotin fill (green) of the fast tibia flexor motor neuron, with nc82 counterstain. Right) Diameters of identified neuron 1) somas, 2) primary neurites, and 3) axons (p<0.01, 2-way ANOVA, Tukey-Kramer correction for multiple comparisons). (**B**) Example traces showing voltage responses to current injection in whole-cell recordings from each motor neuron type. (**C**) Average resting membrane potential for fast (n = 15), intermediate (n = 11), and slow tibia flexor motor neurons (n = 14) (p<0.001, 2-way ANOVA, Tukey-Kramer correction for multiple comparisons). (**D**) Average spontaneous firing rate in fast (n = 15), intermediate (n = 11), and slow neurons (n = 14) (p<$10^{-5}$). (**E**) Input resistance in fast (n = 15), intermediate (n = 11), and slow neurons (n = 14) (p<0.001, 2-way ANOVA, Tukey-Kramer correction for multiple comparisons). All intrinsic properties in C-E) were calculated from periods when the fly was not moving.

was 150 MΩ for fast motor neurons, 300 MΩ for intermediate motor neurons, and 700 MΩ for slow motor neurons. Current injection in the fast and intermediate neurons failed to reliably trigger action potentials, based on recordings from both the cell body (whole-cell) and axon (EMG). This is likely because of intrinsic morphological or electrophysiological properties that electrically isolate the soma from the spike initiation zone (*Sasaki and Burrows, 1998*). By contrast, injecting current into the slow neuron effectively modulated the spike rate (*Figure 3B*).

Overall, our electrophysiological characterization of tibia flexor motor neurons (*Figures 2*, *3*) revealed a systematic relationship between motor neuron anatomy, input resistance, resting membrane potential, and spontaneous firing rate. These properties accompany differences in the force generated by a spike in each neuron, which we quantify next.

## A gradient of force per spike across motor neurons

How much force does a single spike in a motor neuron produce? How does force-per-spike, or *gain*, vary across neurons within a motor pool? To answer these questions, we measured probe displacement as a function of firing rate in each motor neuron type. To evoke spikes in the fast and intermediate neurons, we optogenetically stimulated motor neuron dendrites in the VNC using expression of Chrimson (*Klapoetke et al., 2014*), a red-shifted channelrhodopsin (*Figure 4A–B*). Brief (10–20 ms) flashes of increasing intensity produced increasing numbers of spikes, corresponding spikes in the EMG, and movement of the tibia detected by small displacements of the force probe (*Video 2*). Aligning probe movement to spike onset showed that a single fast motor neuron spike produced ~10 µN of force, resulting in a 50 µm movement of the force probe (*Figure 4A*). An intermediate neuron spike produced ~1 µN that moved the tibia 5 µm (*Figure 4B*). For comparison, during take-off, the peak force production of the fly leg is ~100 µN (*Zumstein et al., 2004*).

Increasing the spike rate of the slow motor neuron with current injection produced small (~1 µm) and slow tibia movements (*Figure 4C*). Decreasing slow motor neuron firing produced tibia extension (*Figure 4C,E*), suggesting that spontaneous firing in this cell contributes to the resting force on the probe. Bath application of 1 µM MLA, an antagonist of nicotinic acetylcholine receptors, led to a decrease in the spontaneous firing rate and reduced the resting force on the probe by ~1.5 µN, or ~15% of the fly's weight (*Figure 4C*). This suggests that the spontaneous firing rate in slow motor neurons is set by excitatory synaptic input.

A comparison of force production as a function of firing rate revealed that fast, intermediate and slow motor neurons occupy distinct force production regimes which span three orders of magnitude (*Figure 4D*). The shapes of these force production curves were also distinct. For fast and intermediate neurons, the force produced by two spikes was ~1.6X the force produced by a single spike (*Figure 4E*) and the force-per-spike curves saturated at ~10 spikes (*Figure 4D*). These observations suggest that the fast and intermediate muscle fibers fatigue during repeated stimulation. In contrast, the resting spike rate of slow motor neurons maintains constant force on the probe (*Figure 4C,F*). Thus, slow motor neurons may be used to maintain body posture, while intermediate and fast motor neurons are transiently recruited to execute body movements. Consistent with this hypothesis, we did not observe that flies maintained sufficient resting force on the probe (>5 s) to require sustained firing of fast or intermediate neurons (*Figure 1E*).

The rates of force generation for the fast and intermediate motor units were greater than the slow motor unit. For example, the effect of a spike in the fast and intermediate motor neurons reached half maximal force in ~8.5 ms (*Figure 4H*), whereas the effect of increasing the spike rate in slow motor neurons was gradual and did not reach its peak within 500 ms. Hyperpolarizing slow motor neurons required ~100 ms to reach its maximal effect on force production, which could reflect slow release of muscle tension or could be due to activity of other motor neurons in the population that influence resting force on the probe.

We recorded from several other motor neurons controlling tibia flexion, none of which had more extreme properties than the fast and slow motor neurons in *Figure 4*. Importantly, these

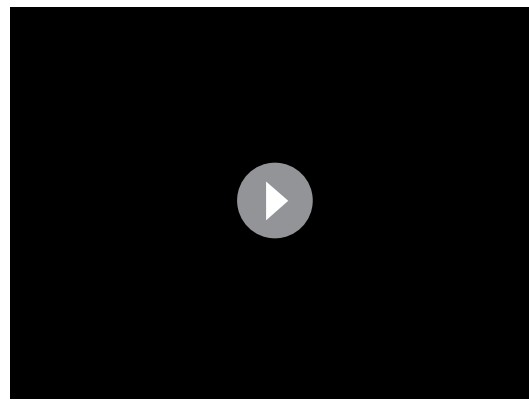

**Video 2.** Motor neuron electrophysiology, force production, and tibia movement. The video introduces 1) the dendritic morphology and axon projection of the three neurons we studied: fast, intermediate and slow; 2) shows video of individual trials from a fast neuron in which one or four spikes are driven with optogenetic stimulation in the VNC, together with probe movement, the whole-cell current clamp recording from the soma, and the EMG record from the leg; 3) shows video of individual trials from an intermediate neuron in which one or four spikes are driven with optogenetic stimulation in the VNC; 4) shows video of a trial from a slow motor neuron in which current injection at the soma depolarizes the neuron and drives ~ 100 spikes, and a trial in which the slow neuron is hyperpolarized, reducing the force on the probe. Videos are slowed 3X.

https://elifesciences.org/articles/56754#video2

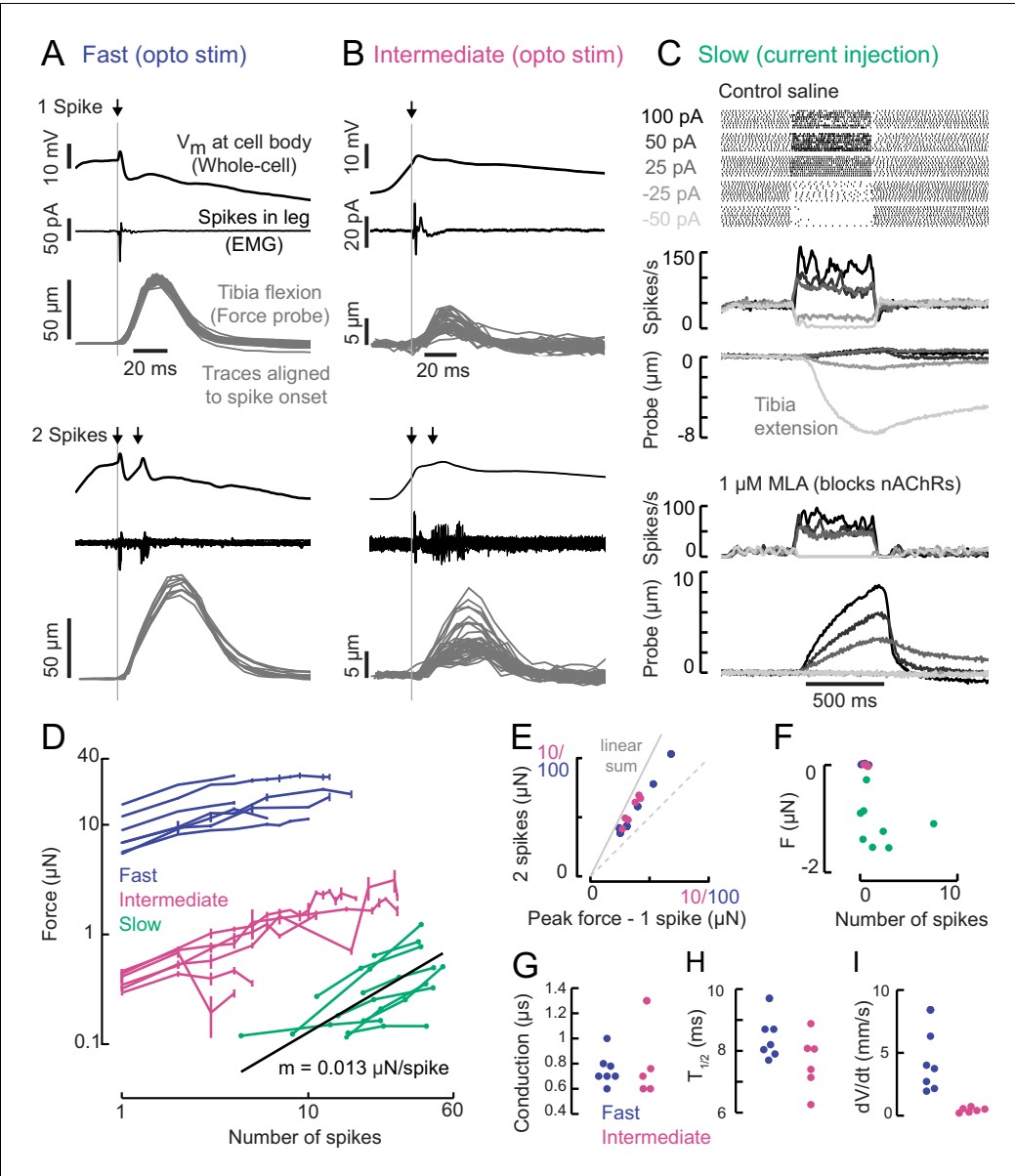

**Figure 4.** A gradient of force production among tibia flexor motor neurons. (**A**) Optogenetic activation of a fast flexor motor neuron expressing CsChrimson (50 ms flash from a 625 nm LED,~2 mW/mm²). Traces show average membrane potential for trials with one (top) and two (bottom) spikes, the average EMG (top) or overlaid EMGs (bottom), and the resulting tibia movement for each trial (50 μm = 11 μN). The jitter in the force probe movement traces results from variability of when a spike occurs relative to the video exposure (170 fps). (**B**) Same as A for an example intermediate flexor motor neuron. (**C**) Tibia movement resulting from current injection in a slow flexor motor neuron. Top: spike rasters from an example cell during current injection. Firing rates are shown below, color coded according to current injection value, followed by the baseline subtracted average movement of the probe (5 μm = 1.1 μN). Bottom: spike rates and probe movement in the presence of the cholinergic antagonist MLA (1 μM), which reduces excitatory synaptic input to the motor neuron. (**D**) Peak average force vs. number of spikes for fast (blue), intermediate (magenta), and slow (green) motor neurons. The number of spikes in slow neurons is computed as the average number of spikes during positive current injection steps minus the baseline firing rate; that is, the number of additional spikes above baseline. The black line is a linear fit to the slow motor neuron data points, with the slope indicated below. (**E**) Peak probe displacement for 2 spikes vs. one spike in fast (blue) and intermediate motor neurons (magenta). (**F**) Summary data showing that zero spikes in fast (n = 7) and intermediate neurons (n = 6) does not cause probe movement, but that hyperpolarization in slow motor neurons (n = 9) causes the fly to let go of the probe, that is, decreases the applied force. The number of spikes (x-axis) is computed as the average number of spikes per trial during the hyperpolarization.(**G**) Delay between a spike in the cell body and

*Figure 4 continued on next page*

*Figure 4 continued*

the EMG spike (conduction delay). Note there may be a delay from the spike initiation zone to the cell body that is not captured (n = 5 intermediate cells), p=0.6, Wilcoxon rank sum test. (**H**) Time to half maximal probe displacement for fast (blue) and intermediate cells (magenta), p=0.2, rank sum test. (**I**) Estimates of the maximum velocity of tibia movement in each fast (blue) and intermediate motor neuron (magenta), p=0.0012, rank sum test. A line was fit to the rising phase of probe points aligned to single spikes as in **B**) and **D**).

The online version of this article includes the following figure supplement(s) for figure 4:

**Figure supplement 1.** Example recordings from other tibia flexor neurons.

cells exhibited similar relationships in morphology, intrinsic properties, and force production (*Figure 4—figure supplement 1*), which provides confidence that these correlations are not an artifact of the motor neurons we have chosen to highlight in this study.

## Motor neurons are recruited in a temporal sequence from weakest to strongest

Calcium imaging from leg muscles suggested that distinct motor units are active during distinct tibia movement regimes, and that this activity follows a recruitment order (*Figure 1*). To test these hypotheses more rigorously, we examined recordings from single cells and pairs of tibia flexor motor neurons during spontaneous leg movements (*Figure 5A–C*).

The membrane potential of each tibia flexor motor neuron reflected probe movement, even during periods of purely subthreshold activity (*Figure 5A–C*). This shows that motor neurons that are not currently spiking are receiving synaptic input that, if shared across the motor pool, could drive other motor neurons to spike (*Gabriel and Büschges, 2007*). This synaptic input could arise from premotor neurons or proprioceptive feedback.

To understand the relationship between motor neuron spiking and tibia movement, we compared force and velocity production in the 25 ms following each motor neuron spike (*Figure 5D–E*). Spikes in each motor neuron type tended to occur during specific regimes of force and velocity. Plots of probe position and movement during example epochs (*Figure 5D*) illustrate that fast motor neurons typically spiked when the tibia was already flexed and moving. In comparison, the only period during which slow motor neurons were silent was when the tibia was extending (negative probe velocity) or fully extended (zero probe position). As a result, the spike-triggered distribution of probe dynamics for the slow motor neuron is broad (*Figure 5E*, right). Large spike rate modulations in the slow motor neuron often coincided with gradual changes in the position of the probe, consistent with the slow kinetics of force production measured using current injection (*Figure 4*). In comparison, spikes in fast motor neurons caused the probe to rapidly accelerate and approach maximum velocity (*Figure 5D–E*). Intermediate neuron spikes could also produce small but measurable increases in force (*Figure 5B*, inset).

Overall, this analysis shows that each motor neuron type has a different regime of force probe position and velocity in which it is likely to spike (*Figure 5—figure supplement 1*). These regimes are overlapping. For example, in the regime where fast motor neuron spikes are likely, both intermediate and slow spikes are likely as well. Similarly, the slow motor neuron is likely to be spiking throughout the intermediate neuron's regime. These relationships between motor neuron spiking and force production are consistent with the existence of a hierarchical recruitment order (*Figure 5F*).

We tested this model by examining paired recordings of somatic (whole-cell) and axonal (EMG) spikes from different motor neurons. Neurons lower in the recruitment hierarchy were consistently active when a neuron producing more force was recruited (*Figure 5G*). In other words, activity in slow neurons preceded that of intermediate neurons, and activity in intermediate neurons preceded that of fast neurons.

Interestingly, we sometimes observed violations of this recruitment order (*Figure 5—figure supplement 1*). The exceptions occurred when the fly was rapidly waving its leg, rather than pulling on the force probe. In these cases, the slow motor neuron spike rate appeared to be suppressed by inhibition. This phenomenon is similar to rapid paw shaking behavior in cats (*Smith et al., 1980*) and rapid swimming in zebrafish (*Menelaou and McLean, 2012*). Such violations suggest that motor

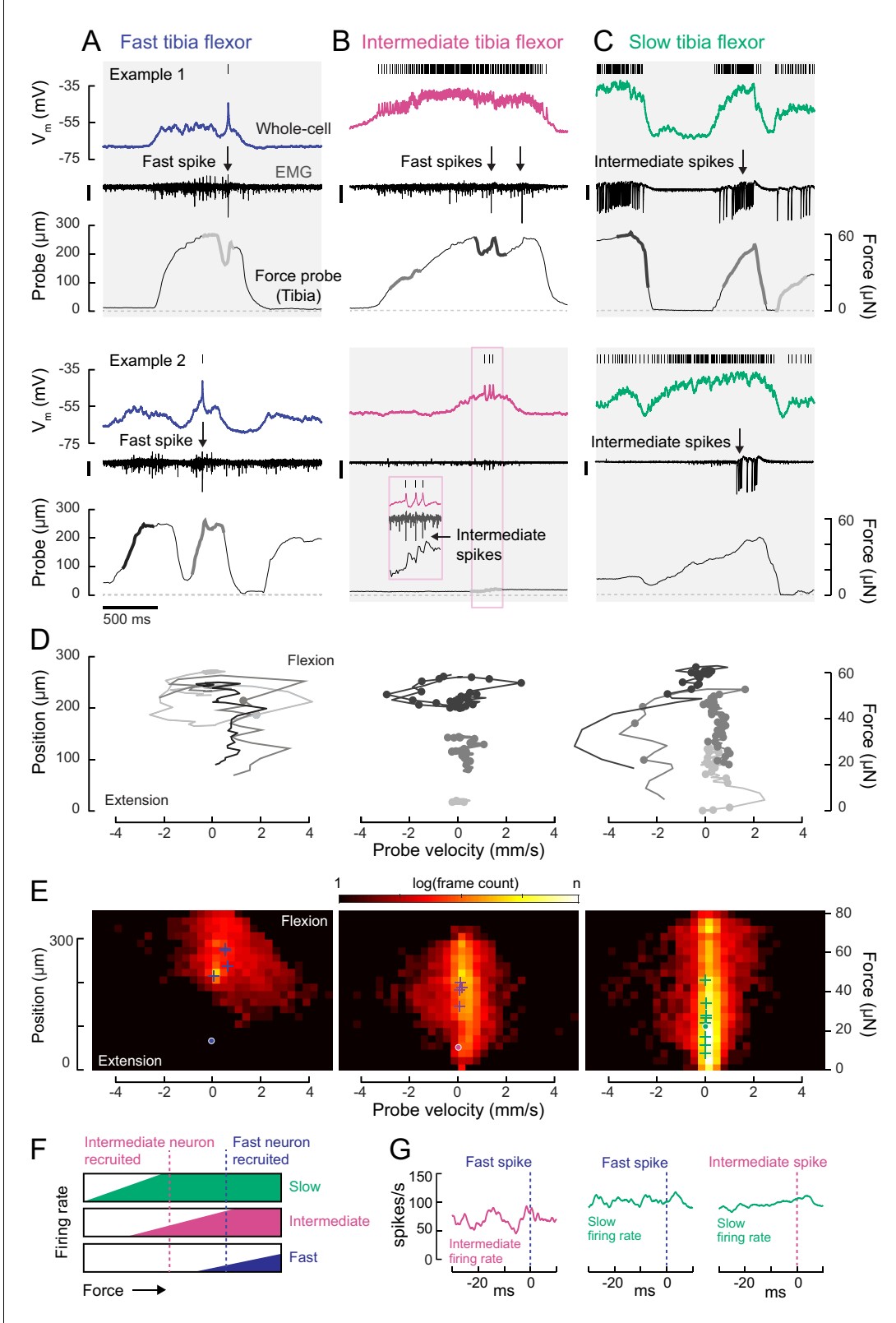

**Figure 5.** Motor neurons are recruited in a specific order across different motor regimes. (A) Membrane potential, EMG, and probe movement, for two example epochs of spontaneous leg movement during a whole-cell recording of the fast motor neuron. Highlighted events are plotted in D. The EMG was recorded in the proximal femur to pick up both fast and intermediate spikes (scale is 10 pA). (B) Same as A, but for an intermediate motor neuron. The EMG for the intermediate neuron was recorded in the proximal part of the femur, the largest events reflect activity in the fast motor neuron (scale

*Figure 5 continued on next page*

*Figure 5 continued*

is 100 pA). (C) Same as A-B, but for a slow motor neuron. The EMG was recorded in the proximal femur, near the terminal bristle, and large events are likely from the intermediate neuron (scale is 100 pA). (D) Example force probe trajectories from movement examples. The grayscale in D matches that of the trajectories in A-C. Each vertex is a single frame; vertices with dots indicate the occurrence of a spike in the whole-cell recording. (E) 2D histograms of probe force and velocity for video frames within 25 ms following a spike across recordings in cells of each type (fast – five neurons, 4666 frames; intermediate – four neurons, 11,833 frames; slow – nine neurons, 65,473 frames). Crosses indicate the centroids of the histograms for individual neurons. Dots outlined in white indicate centroid of the 2D histograms of all frames, within and outside the 25 ms window. Color scale shows the log number of frames in each bin, normalized to the total number of frames. (F) Schematic illustrating firing rate predictions of the recruitment hierarchy. As force increases, left to right, first the slow neuron begins firing, then the intermediate neuron (magenta line). Once the fast neuron spikes, both intermediate and slow neurons are already firing (blue dotted line). (G) Left) Spike rate of intermediate neuron in 30 ms preceding fast EMG spike (n = 4 pairs); center) slow neuron spike rate preceding fast EMG spike (n = 2 pairs); right) and slow neuron spike rate preceding intermediate EMG spike (n = 3 pairs).

The online version of this article includes the following figure supplement(s) for figure 5:

**Figure supplement 1.** Testing the recruitment hierarchy of motor neurons, in paired recordings and as a function of force probe position and velocity.

neuron excitability is not the only factor that determines recruitment order. Rather, differences in presynaptic input may selectively de-recruit slow motor neurons through inhibition.

## A gradient of sensory input to motor neurons

So far, we have shown that tibia flexor motor neurons exhibit a gradient of functional properties (*Figures 2–4*) and a recruitment order during natural leg movements (*Figure 5*). We next asked how motor neurons respond to proprioceptive sensory feedback. A key function of proprioceptive feedback is to maintain stability by counteracting sudden perturbations, such as when an animal stumbles during walking (*Tuthill and Azim, 2018*). Larger perturbations require more corrective force, so as feedback increases it should result in the recruitment of additional motor neurons with increasing force production capacity.

We first compared the amplitude and dynamics of proprioceptive feedback in flexor motor neurons in response to ramping movements of the tibia (*Figure 6A–B*). Passive extension of the tibia caused excitatory postsynaptic potentials (PSPs) in all three flexor motor neurons, a response known as a resistance reflex. PSP amplitude was largest in slow motor neurons and smallest in fast neurons (*Figure 6C–D*). The slope of the rising phase of the PSP increased with the rate of extension. In the fast and intermediate neurons, sensory-evoked PSPs typically failed to elicit spikes, whereas the firing rate of slow motor neurons was significantly modulated. Slow motor neurons were exceptionally sensitive to passive tibia movement: a 1° change in tibia angle resulted in a significant change in firing rate (*Figure 6F*). Extremely fast extension movements were sufficient to evoke single spikes in intermediate neurons (*Figure 6—figure supplement 1*), but we never observed feedback-evoked spikes in fast motor neurons. The amplitude of the PSP evoked by proprioceptive feedback did not vary across different resting femur-tibia joint angles (*Figure 6—figure supplement 1*).

In addition to differences in amplitude, the sign and dynamics of proprioceptive feedback varied across fast, intermediate, and slow motor neurons. Fast and slow motor neurons both hyperpolarized slightly at flexion onset, while intermediate neurons depolarized once the flexion event ceased (*Figure 6E*). These results demonstrate that the different tibia flexor motor neurons receive distinct feedback signals from leg proprioceptors.

As reported in recordings from other insect species (*Bässler and Büschges, 1998*), we occasionally observed a reflex reversal, in which the sign of the proprioceptive PSP reversed (*Figure 6—figure supplement 1*). In most instances, tibia extension led to excitatory responses in flexor motor neurons. But occasionally, typically when the fly was actively moving as the stimulus was delivered, extension produced an inhibitory PSP. Reflex reversal is an important mechanism that could allow the fly to switch between active and passive motor states (*Bässler and Büschges, 1998*; *Tuthill and Wilson, 2016a*).

A major source of proprioceptive information about tibia position is the femoral chordotonal organ (FeCO). The *Drosophila* FeCO is comprised of ~150 mechanosensory neurons that encode position, movement, and vibration of the femur-tibia joint (*Mamiya et al., 2018*). We used optogenetic activation (*iav-LexA > Chrimson*) to measure proprioceptive feedback from the FeCO to each motor neuron type (*Figure 6—figure supplement 2*). Stimulating FeCO neurons caused the fly to

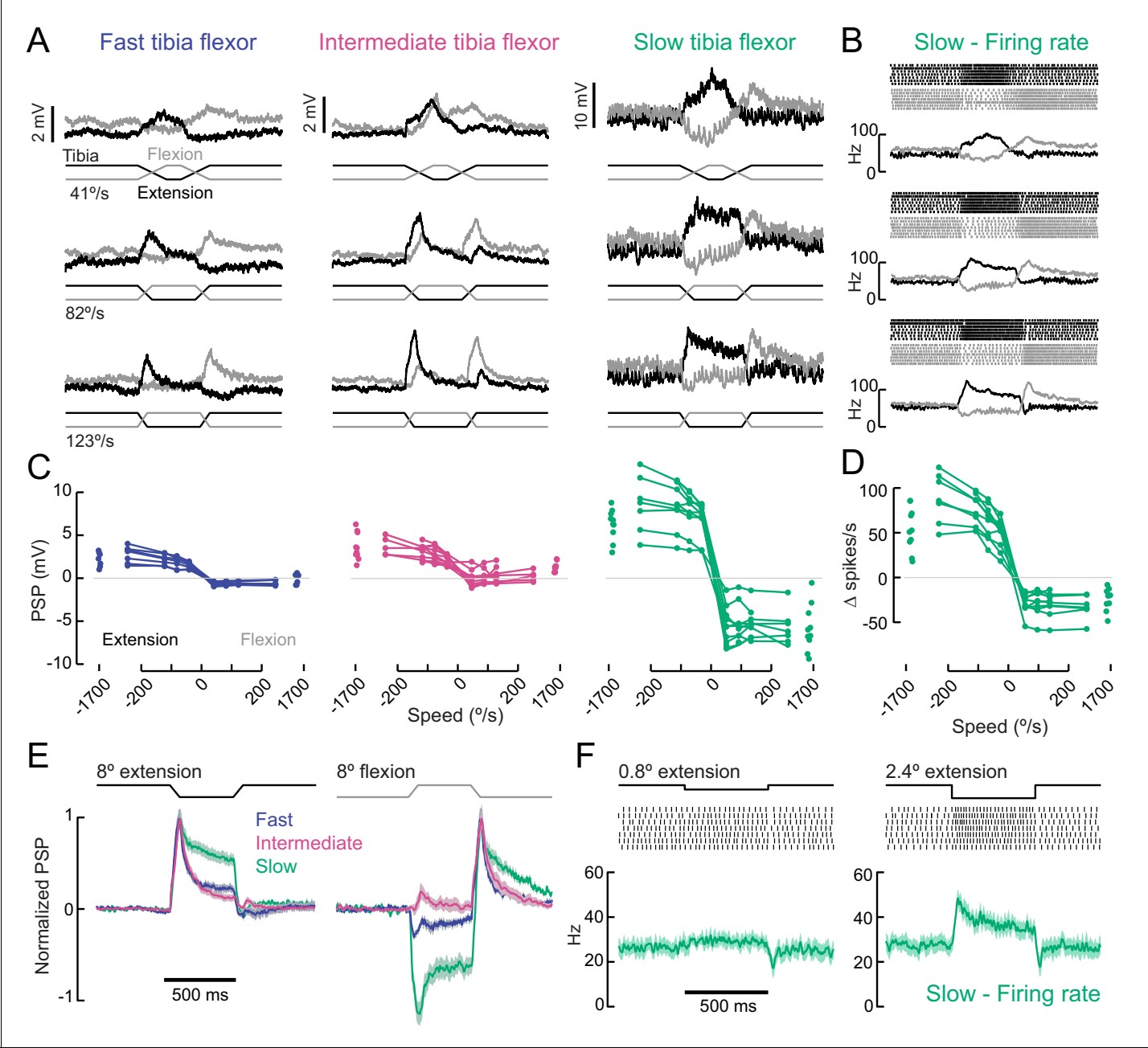

**Figure 6.** A gradient of proprioceptive feedback to motor neurons controlling tibia flexion. (**A**) Whole-cell recordings of fast, intermediate, and slow motor neurons in response to flexion (black) and extension (gray) of the tibia. Each trace shows the average membrane potential of a single neuron (n = 7 trials). A 60 μm movement of the probe resulted in an 8° change in femur-tibia angle. (**B**) Spike rasters and firing rates for the same slow neuron as in A. (**C**) Amplitude of the postsynaptic potential (PSP) vs. angular velocity of the femur-tibia angle (n = 7 fast, seven intermediate, nine slow cells). Responses to fast extension (−123°/s) are not significantly different for fast and intermediate neurons (p=0.5) but are different for slow neurons (p<10⁻⁵, 2-way ANOVA, Tukey-Kramer correction for multiple comparisons). (**D**) Amplitude of firing rate changes in slow motor neurons vs. angular velocity of the femur-tibia angle. (**E**) Time-course of normalized average responses (± s.e.m. of baseline subtracted responses) to tibia extension (left) and flexion (right, n = 7 fast, seven intermediate, nine slow cells). (**F**) Sensitivity of firing rate to small tibia movements of 6 μm (0.8°) or 18 μm (2.4°). Spike rasters are from representative cells, the average spike rate across cells is plotted in green below.

The online version of this article includes the following figure supplement(s) for figure 6:

**Figure supplement 1.** Properties of sensory feedback to motor neurons.
**Figure supplement 2.** Optogenetic activation of proprioceptive sensorimotor circuits.

extend and then flex its tibia in a rapid and stereotyped manner. Motor neuron responses to FeCO stimulation increased from slow motor neurons to fast motor neurons, consistent with responses to passive stimulation. In interpreting these experiments, it is important to note that Chrimson expression in FeCO neurons reduced motor neuron responses to passive leg movements compared to control flies (*Figure 6—figure supplement 2*). We speculate that this side-effect of Chrimson expression is caused by a homeostatic decrease in the gain of proprioceptive feedback pathways following increased excitability.

Overall, these data demonstrate that the FeCO provides proprioceptive feedback to tibia flexor motor neurons, and that the amplitude and dynamics of proprioceptive feedback signals vary across the different motor neurons. This gradient of sensory feedback amplitude is likely shaped by the concurrent gradient of motor neuron intrinsic properties (*Figure 3*). We observed differences in the sign and dynamics of proprioceptive feedback, which indicate that tibia flexor motor neurons receive distinct presynaptic inputs. These differences also suggest that proprioceptive feedback signals may be specialized for controlling particular motor neurons.

## Optogenetic perturbation of single motor neurons alters walking behavior

We have thus far measured force production and recruitment order of slow, intermediate, and fast tibia flexor motor neurons under controlled conditions. We next used optogenetics to alter motor neuron activity in tethered, walking flies, in order to test how sensorimotor feedback loops respond to perturbations of normal activity patterns. Flies flex the tibia of the front leg to pull the body forward during walking (Video 4), so we examined how each motor neuron contributes to this movement. We focused a green laser at the ventral thorax, at the base of the left front (T1) leg to optogenetically activate (*Gal4 >CsChrimson*) (*Klapoetke et al., 2014*) or silence (*Gal4 >gtACR1*) (*Mohammad et al., 2017*) each motor neuron type. As a control, we used an *empty-Gal4* driver line, which has the same genetic background but lacks Gal4 expression.

To verify our optogenetic manipulations and characterize basic leg reflexes, we first measured the movement of the femur-tibia joint in headless flies with their legs unloaded (i.e., the fly was suspended in air, *Figure 7A*). Activation of fast motor neurons caused rapid flexion of the tibia (*Figure 7B,C*, *Video 3*). We observed repeated patterns of tibia flexion and extension, suggesting that the fast neuron fired single spikes and that the resulting flexion was immediately countered by a resistance reflex. Activation of intermediate neurons caused slower tibia flexion, while activation of slow neurons caused even slower contractions, often followed by unexpected, prolonged tibia extension. Silencing each motor neuron type did not consistently evoke leg movements, though silencing the full motor neuron population resulted in paralysis (*Video 4*). These results are consistent with our measures of force production during electrophysiology (*Figures 4*, *5*) and provide a useful baseline for interpreting results from walking flies.

To optogenetically manipulate motor neuron activity during walking, we positioned tethered flies on a spherical treadmill (*i.e.*, a Styrofoam ball) within a visual arena (*Reiser and Dickinson, 2008*; *Figure 7D*). Previous studies have found that walking speed and gait on the treadmill are similar to freely walking flies (*Szczecinski et al., 2018*). In our setup, the fly was motivated to walk forward with a wiggling stripe (*Figure 7E*), and we tracked the treadmill and the fly's behavior with multiple high-speed cameras.

We first asked how flies would respond to transient optogenetic activation of the different tibia flexor motor neurons. Activation of the fast motor neuron caused the tibia of the front leg to persistently flex, thus, interrupting walking (*Figure 7E*, *Video 5*). Consequently, flies appeared to slow down, as measured by a decrease in forward walking velocity. Activation of the slow motor neurons also interrupted walking (*Figure 7E*), but this was because the fly reacted by extending its tibia, as when the legs were unloaded (*Video 7*). Based on the muscle fibers innervated by the slow motor neuron, we speculate that optogenetic activation of this neuron may increase twisting at the joint, which the fly reacts to with a compensatory extension of the leg. Unlike the other two cells, activation of the intermediate motor neuron appeared to cause the flies to transiently increase their walking speed. Activation of the intermediate motor neuron in stationary flies also caused them to start walking (*Figure 7E–F*, *Video 6*). Extra spikes in the intermediate tibia flexor motor neuron may accelerate the rotation of the treadmill, forcing the other legs to move faster in response. Overall,

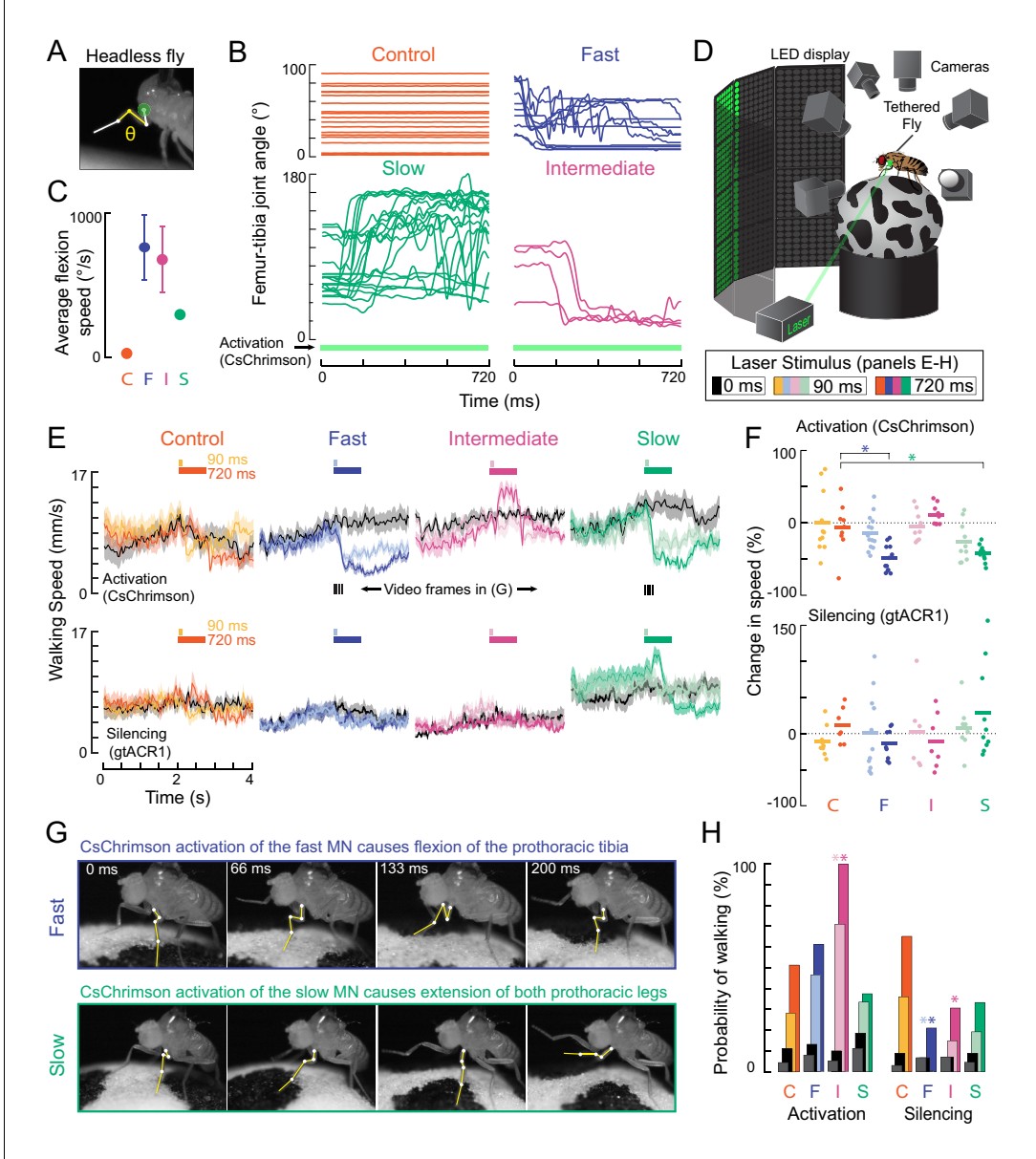

**Figure 7.** Optogenetic perturbation of motor neurons in behaving flies. (**A**) Example frame illustrating behavioral effects of optogenetically activating leg motor neurons in headless flies. A green laser (530 nm) is focused at the coxa-body joint of the fly's left front leg (outlined in white) and the femur-tibia joint (yellow) is monitored with high-speed video. (**B**) Example traces of femur-tibia joint angles during optogenetic activation (*Gal4 >CsChrimson*) of each motor neuron type in headless flies. The control line has the same genetic background as motor neuron lines but lacks Gal4 expression (*BDP-Gal4 >CsChrimson*).( **C**) Average (± sem) tibia flexion speed during the first 300 ms of CsChrimson activation (n = 5 flies of each genotype). (**D**) Schematic of the behavioral setup, in which a tethered fly walks on a spherical treadmill. The treadmill and fly are tracked with separate cameras. As in **A**), a green laser is focused on the coxa of the left front leg. (**E**) Average treadmill forward velocity (± sem) of walking flies, when activating (top row) or silencing (bottom row) control (n = 11, 13), fast (n = 15, 14, intermediate (n =9,12), or slow (n = 11, 13) motor neurons for 90 ms (dark colors) or 720 ms (light colors). Black traces indicate trials with no laser (0 ms). Gray boxes indicate the period of optogenetic stimulation. Black dashes above fast and slow flexor activation traces indicate the time points in panel G. (**F**) Percent change (change in speed/pre-stimulus speed) in the average running speed of flies shown in C during stimulation+200 ms for activation (top) and silencing (bottom). Asterisks indicate p<0.01 relative to control (bootstrapping with false discovery rate correction; see methods for details). (**G**) Example frames showing representative behavioral responses of running flies during the first 200 ms of fast (top) and slow (bottom) MN activation. (**H**) Probability that a stationary fly initiated a walking bout during the 500 ms following either MN activation (left) or silencing (right). Dark colors are for the 90 ms stimulus (or no stimulus – black) and light colors are for the 720 ms stimulus (or no stimulus – gray). Asterisks indicate p<0.01 relative to control (bootstrapping with false discovery rate correction; see Methods for details). The online version of this article includes the following figure supplement(s) for figure 7:

**Figure supplement 1.** Effects of tibia motor neuron silencing and activation on leg kinematics.

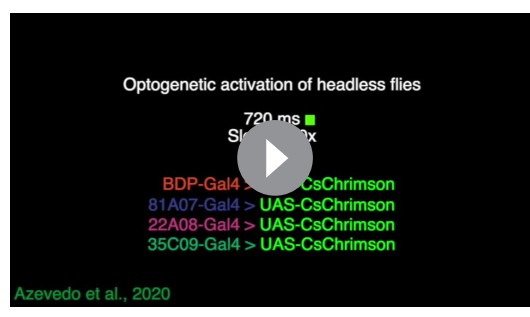

**Video 3.** Optogenetic activation of motor neurons in headless flies. The video shows the movements of the flies left front tibia caused by optogenetic activation of the fast, intermediate and slow neurons and a control line (*BDP-Gal4*). The video shows 1) the tibia flexion during the entire 720 ms laser stimulation period slowed 10X; 2) the first 300 ms of the same event, slowed 40X.

https://elifesciences.org/articles/56754#video3

these results show that activation of different motor neurons drive distinct behavioral responses that depend on the fly's behavioral state (*e.g.*, walking vs. standing).

Optogenetically silencing motor neurons had less pronounced effects on walking speed (*Figure 7E*). However, silencing of fast and intermediate neurons decreased the probability that a stationary fly would initiate walking compared to controls (*Figure 7H*, *Videos 6* and *7*). These results suggest that motor neurons with a high force production capacity (*Figure 4*) may be dispensable for normal walking behavior, but their activity may be important for producing the high levels of force required to initiate walking.

## Discussion

In this study, we characterized the cellular morphology, electrical excitability, force production, and proprioceptive feedback among motor neurons that control flexion of the *Drosophila* tibia. We discovered a gradient of properties that co-varies across the motor pool (*Figures 1*, *2*). At one end of the spectrum is the fast motor neuron, which has a larger diameter axon and dendrites, low input resistance, weak sensory input, and produces enough force with each spike to support the fly's entire body weight (*Figures 3*, *4*). At the other end, the slow motor neuron has fine axons and dendrites and high input resistance; each spike in the slow motor neuron produces <1% the force of a fast motor neuron spike. The slow motor neuron also has a high spontaneous firing rate that maintains resting muscle tension (*Figure 4*), and it receives strong proprioceptive feedback that continuously modulates its firing rate (*Figure 6*).

We found that leg motor neurons are recruited in a specific order: small, low gain motor neurons that generate weak forces are recruited first, and larger, more powerful motor neurons are recruited later (*Figure 5*). Consistent with this recruitment order, proprioceptive feedback has the greatest effect on the slow motor neuron: tibia movements of less than 1° significantly impact slow motor neuron firing (*Figure 6*). By comparison, proprioceptive feedback has a much smaller effect on the firing rate of intermediate and fast motor

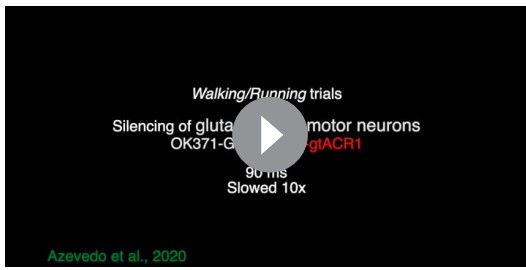

**Video 4.** Optogenetic silencing in all motor neurons controlling the front leg. Motor neurons labelled by *OK371-Gal4* expressed gtACR1. The video shows the behavior of different flies on different trials while 1) flies were walking during a 90 ms laser stimulus; 2) flies were stationary during a 90 ms laser stimulus; 3) flies were walking during a 720 ms laser stimulus; 4) flies were stationary during a 720 ms laser stimulus. Videos of behavior are slowed 10X.

https://elifesciences.org/articles/56754#video4

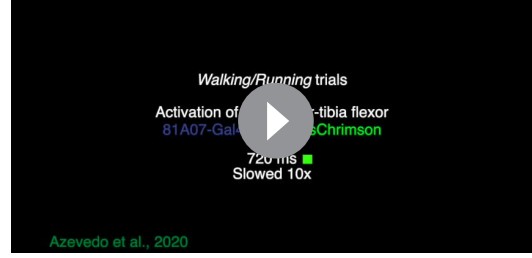

**Video 5.** Optogenetic activation and silencing of fast motor neurons in behaving flies. The video shows the behavior of different flies on different trials while 1) flies were walking and the fast neuron expressed CsChrimson; 2) flies were stationary, and the fast neuron expressed CsChrimson; 4) flies were walking, and the fast neuron expressed gtACR1; 3) flies were stationary, and the fast neuron expressed gtACR1. Videos of behavior are slowed 10X.

https://elifesciences.org/articles/56754#video5

neurons. Fly behavior is also quite sensitive to aberrant optogenetic recruitment of motor neurons (*Figure 7*). For example, flies fail to initiate walking if unable to recruit slow motor neurons at the bottom of the recruitment hierarchy.

A relationship between force production and recruitment order is a common feature of vertebrate motor systems (*Henneman et al., 1965a*; *Henneman and Olson, 1965*; *Kernell, 2006*; *Mcphedran et al., 1965*; *McPhedran et al., 1965*). This organization has been proposed to confer a number of computational and energetic advantages. Recruitment of additional motor units increases force nonlinearly, overcoming suppressive nonlinearities in spike rates and muscle force production (*Bakels and Kernell, 1994*; *Monster and Chan, 1977*). Furthermore, the relative increase in force does not decrease with successive recruitment, as it would if all motor units produced similar amounts of force. Thus, much like Weber's law describes the constant sensitivity to relative stimulus intensity (*Fechner, 1860*), a recruitment hierarchy maximizes the resolution of motor unit force while also simplifying the dimensionality of the motor system (*Henneman et al., 1965b*; *Mendell, 2005*).

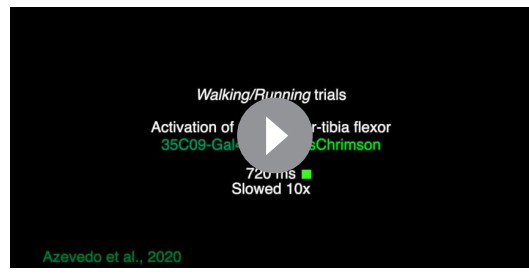

**Video 7.** Optogenetic activation and silencing of slow motor neurons in behaving flies. The video shows the behavior of different flies on different trials while 1) flies were walking and the slow neuron expressed CsChrimson; 2) flies were stationary, and the slow neuron expressed CsChrimson; 3) flies were walking, and the slow neuron expressed gtACR1; 4) flies were stationary, and the slow neuron expressed gtACR1. Videos of behavior are slowed 10X.
https://elifesciences.org/articles/56754#video7

## Violations of the size principle and their implications for premotor circuit organization

A key assumption of the original size principle is that all motor neurons within a pool receive the same presynaptic input (*Henneman et al., 1965a*); the alternative is that different premotor neurons provide input to different subsets of motor neurons. Our results suggest the motor system controlling the fly tibia operates in a middle ground between these two extremes. Although tibia motor neurons generally follow a recruitment order in accordance with the size principle (*Figure 5*), we found that the dynamics of proprioceptive feedback vary across motor neurons within the pool (*Figure 6*). We also observed a small but significant proportion of behavioral events in which recruitment order was violated (*Figure 5—figure supplement 1*). These data suggest that although the tibia flexor motor neuron pool may share some presynaptic inputs, they are not identical. Thus, motor control of the fly tibia is more complex than a straightforward implementation of the size principle.

Similar exceptions to the strict dogma of the size principle have been observed in vertebrate species. For instance, the effect of proprioceptive feedback varies across a pool of motor neurons controlling the cat leg (*Heckman and Binder, 1988*; *Heckman and Binder, 1991*). During rapid escape behaviors in zebrafish, slow motor neurons can completely drop out of the population firing pattern (*Menelaou and McLean, 2012*). Muscles controlling human fingers can change recruitment order based on movement direction (*Desmedt and Godaux, 1977*). These examples suggest that some behaviors exhibit more degrees of freedom than

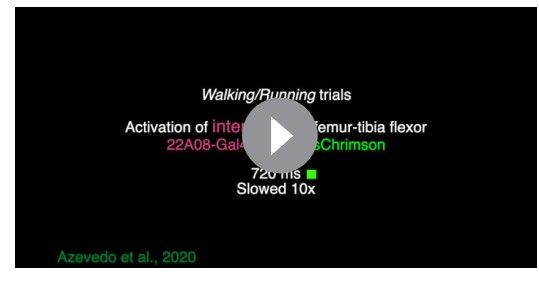

**Video 6.** Optogenetic activation and silencing of intermediate motor neurons in behaving flies. The video shows the behavior of different flies on different trials while 1) flies were walking and the intermediate neuron expressed CsChrimson; 2) flies were stationary, and the intermediate neuron expressed CsChrimson; 3) flies were walking, and the intermediate neuron expressed gtACR1; 4) flies were stationary, and the intermediate neuron expressed gtACR1. Videos of behavior are slowed 10X.
https://elifesciences.org/articles/56754#video6

can be supported by recruitment of motor neurons based on their intrinsic excitability alone.

Investigations of the vertebrate spinal cord have also identified circuit motifs that support flexible control of motor neurons within a pool (*Kiehn, 2016*; *McLean and Dougherty, 2015*). For example, recordings in turtles and zebrafish have shown that motor neurons receive coincident excitation and inhibition, which could underlie selective de-recruitment (*Berg et al., 2007*; *Kishore et al., 2014*). Zebrafish spinal cord premotor neurons provide patterned inhibition to speed-specific circuits (*Callahan et al., 2019*; *Menelaou et al., 2019*) and can flexibly switch between different motor neuron recruitment patterns (*Bagnall and McLean, 2014*).

Our characterization of identified tibia motor neurons provides a handle to investigate similar premotor circuit motifs for flexible limb motor control in the fly. Little is currently known about the leg premotor circuitry in *Drosophila*. However, the hypotheses generated by this work should soon be testable using connectomic reconstruction of identified motor neurons and their presynaptic inputs in the *Drosophila* VNC (*Maniates-Selvin et al., 2020*).

## Organization of motor pools controlling the *Drosophila* tibia

The muscles that control flexion of the *Drosophila* tibia are innervated by approximately 15 motor neurons (*Baek and Mann, 2009*; *Brierley et al., 2012*). Here, we chose to analyze three identified neurons that span the extremes of this motor pool. Each of these motor neurons was stereotyped across flies in its anatomy, physiology, and function. The fast tibia flexor motor neuron labeled by *R81A07-Gal4* is the only motor neuron that innervates the large tibia flexor muscle fibers in the middle of the femur (*Figure 2B*). Because it has the largest axon of any motor neuron in the femur, the fast motor neuron also produces the largest extracellular spikes (*Figure 2D*). The slow tibia flexor motor neuron labeled by *R35C09-Gal4* is one of 8–9 motor neurons that innervates tibia flexor muscle fibers located at the distal tip of the femur. Measurements of force production from other motor neurons that innervated this distal region (*Figure 4—figure supplement 1*) suggest that the *R35C09-Gal4* neuron is among the weakest within this group. Finally, we studied an intermediate motor neuron labeled by *R22A08-Gal4* that innervates muscle fibers separate from the fast and slow neurons. Previous anatomical studies suggest that 2–5 neurons innervate nearby muscle fibers in the proximal region of the femur. Our recordings of other tibia flexor motor neurons (*Figure 4—figure supplement 1*) were consistent with the gradient of properties we describe in detail for three identified motor neurons, which suggests that the relationship between anatomy, physiology, and force production applies to other neurons in the tibia flexor motor pool.

The structure of the leg motor system in *Drosophila* has several similarities to other well-studied walking insects. In the metathoracic leg of the locust, a single tibia flexor muscle is innervated by nine motor neurons that were also classified into three groups: fast, intermediate, and slow (*Burrows and Hoyle, 1973*; *Phillips, 1981*; *Sasaki and Burrows, 1998*). Different motor neurons within this pool lie along a gradient of intrinsic properties (*Sasaki and Burrows, 1998*) and are sensitive to different types of proprioceptive feedback, such as position vs. velocity or fast vs. slow movements (*Field and Burrows, 1982*; *Newland and Kondoh, 1997*). The femur of the stick insect is innervated by flexor motor neurons that were also described as slow, semi-fast, and fast, based on their intrinsic properties and firing patterns during behavior (*Bässler, 1993*; *Schmidt et al., 2001*).

Much of the work in bigger insects and crustaceans has focused on the most reliably identifiable neurons, the fast and slow extensor tibiae (FETi and SETi), antagonists to the more diverse tibia flexor motor neurons. Consequently, details of how sensory feedback and local interneurons recruit specific subsets of flexor motor neurons has been relatively understudied (*Clarac et al., 2000*) (but see *Gabriel et al., 2003*; *Gabriel and Büschges, 2007*; *Hill and Cattaert, 2008*). Our work exemplifies the advantage of using *Drosophila* genetics to identify cell types, and even individual cells, within a diverse motor pool. Genetic markers have also recently been identified for tibia extensor motor neurons in flies (*Venkatasubramanian et al., 2019*), which will enable future investigation of premotor input to antagonist motor neurons.

Flies differ from these other invertebrates in one major respect: while most arthropods possess GABAergic motor neurons that directly inhibit leg muscles, holometabolous insects such as *Drosophila* do not (*Schmid et al., 1999*; *Witten and Truman, 1998*). Indeed, we found that a transgenic line for GABAergic neurons (*Gad1-Gal4*; *Diao et al., 2015*) does not label any axons in the *Drosophila* leg (data not shown). The presence of inhibitory motor neurons has been proposed as a key underlying reason why insects have been able to achieve flexible motor control with small numbers of motor

neurons (*Belanger, 2005*; *Wolf, 2014*). That flies lack this capability means that other mechanisms must be at play. Fly legs are innervated by neurons that release neuromodulators, such as octopamine, which could underlie flexible tuning of muscle excitability (*Zumstein et al., 2004*).

## Behavioral function of different motor neurons within a pool

The large range of gain, or force production, at the femur-tibia joint implies that different subgroups of flexor neurons will be recruited during distinct behaviors. Activity in slow motor neurons produces small, low force movements, while fast motor neuron activity produces fast, high amplitude movements. Consistent with this division, the output of slow motor neurons is more strongly influenced by feedback from leg proprioceptors (*Figure 6*). Thus, proprioceptive feedback may continuously modulate the firing rate of slow motor neurons for precise stabilization of posture during standing or grooming. When the slow motor neuron is optogenetically activated (*Figure 7*), flies stop walking and extend their legs. The reason for this response is unclear but could reflect the fly's reaction to a loss of autonomous motor control.

In contrast to the postural movements controlled by slow motor neurons, rapid, stereotyped movements like escape (*Card and Dickinson, 2008*; *Zumstein et al., 2004*) are likely to use fast motor neurons whose activity is less dependent on sensory feedback. This division of labor may also provide energy efficiency: slow contracting muscle fibers use aerobic metabolism and take advantage of energy stores in the form of glycogen, while fast muscle fibers are anaerobic and lack energy stores, leading to more rapid fatigue (*Kernell, 2006*).

Even for a specific behavior, such as walking, different motor neurons may be recruited in different environments and contexts. We saw that optogenetically activating intermediate motor neurons caused stationary flies to start walking and walking flies to walk faster (*Figure 7*). In a similar manner, stick insects walking on treadmills typically recruit fast motor neurons only during fast walking and, in that case, late in the stance phase. But as friction on the treadmill increases, fast neurons are recruited earlier during stance (*Gabriel et al., 2003*). Leg kinematics and force production also change as insects walk up or down inclines (*Dallmann et al., 2019*; *Wöhrl et al., 2017*). When locusts (*Duch and Pflüger, 1995*) and cockroaches (*Larsen et al., 1995*; *Watson and Ritzmann, 1997*) are forced to walk upside-down, fast flexor neurons are recruited to allow the animal to grip the substrate. This context-dependence is not limited to walking: small changes in body posture can have a large effect on which motor neurons are recruited during target reaching movements in locusts (*Page et al., 2008*).

In this study, we chose to focus on flexion of the femur-tibia joint of the fly's front leg. How might these results compare to motor neuron pools of antagonist muscles? Tibia flexor motor neurons outnumber the extensor neurons (*Baek and Mann, 2009*; *Brierley et al., 2012*), and thus are likely to possess a shallower gradient of intrinsic and functional properties, as has been found in crayfish (*Hill and Cattaert, 2008*). Invertebrate muscles also exhibit polyneural innervation, such that a given muscle fiber may be innervated by multiple motor neurons (*Brierley et al., 2012*; *Hoyle, 1983*; *Sasaki and Burrows, 1998*). Polyneural innervation could allow independent activation of motor neurons in order to use the same muscle fibers during different contexts, or it could make the force produced by one motor neuron dependent on coincident activity in another motor neuron (*Sasaki and Burrows, 1998*).

## Summary

In this study, we describe an organizing architecture for motor control of tibia movement in *Drosophila*. These neurons constitute the output of motor circuits that flexibly control a wide range of fly behaviors, from walking to aggression to courtship. Studies of motor control in the fly VNC are now possible thanks to an increasing catalog of genetically-identified cell types (*Lacin et al., 2019*; *Namiki et al., 2018*; *Shepherd et al., 2019*), connectomic reconstruction with serial-section EM (*Maniates-Selvin et al., 2020*) and the ability to image from VNC neurons in walking flies (*Chen et al., 2018*). The leg motor neurons we describe also provide valuable targets for understanding the coordinated development of motor neuron and muscle properties (*Enriquez et al., 2015*; *Venkatasubramanian et al., 2019*). We anticipate that *Drosophila* will be a useful complement to other model organisms in understanding the neural basis of flexible motor control.

# Materials and methods

## Key resources table

| Reagent type (species) or resource | Designation | Source or reference | Identifiers | Additional information |
|---|---|---|---|---|
| Genetic reagent (*D. melanogaster*) | 'w[*]; P{w[+mC]=Mhc-GAL4.K}2' | Bloomington *Drosophila* Stock Center | RRID:BDSC_55133 | |
| Genetic reagent (*D. melanogaster*) | 'P{y[+t7.7] w[+mC]=20XUAS-IVS-mCD8::GFP}attP2' | Bloomington *Drosophila* Stock Center | RRID:BDSC_32194 | |
| Genetic reagent (*D. melanogaster*) | 'P{y[+t7.7] w[+mC]=GMR22A08-GAL4}attP2' | Bloomington *Drosophila* Stock Center | RRID:BDSC_47902 | |
| Genetic reagent (*D. melanogaster*) | 'P{JFRC7-20XUAS-IVS-mCD8::GFP} attp40' | Other | FBrf0212432 | Barret Pfeiffer, Janelia Farm, HHMI |
| Genetic reagent (*D. melanogaster*) | 'MHC-LexA' | Other | | Richard Mann, Columbia University |
| Genetic reagent (*D. melanogaster*) | '13XLexAop2-IVS-GCaMP6f-p10}su(Hw)attP5' | Bloomington *Drosophila* Stock Center | RRID:BDSC_44277 | |
| Genetic reagent (*D. melanogaster*) | 'Berlin-K' | Bloomington *Drosophila* Stock Center | RRID:BDSC_8522 | |
| Genetic reagent (*D. melanogaster*) | 'P{y[+t7.7] w[+mC]=GMR81A07-GAL4}attP2' | Bloomington *Drosophila* Stock Center | RRID:BDSC_40100 | |
| Genetic reagent (*D. melanogaster*) | 'P{y[+t7.7] w[+mC] =GMR35 C09-GAL4}attP2' | Bloomington *Drosophila* Stock Center | RRID:BDSC_49901 | |
| Genetic reagent (*D. melanogaster*) | '10XUAS-syn21-Chrimson88-tDT3.1 (attP18)' | DOI: 10.1016/j.neuron.2017.03.010 | | Michael Reiser, Janelia Farm, HHMI |
| Genetic reagent (*D. melanogaster*) | 'P{y[+t7.7] w[+mC] =20 XUAS-IVS-CsChrimson.mVenus}attP2' | Bloomington *Drosophila* Stock Center | RRID:BDSC_55136 | |
| Genetic reagent (*D. melanogaster*) | 'iav-LexA' | Bloomington *Drosophila* Stock Center | RRID:BDSC_52246 | |
| Genetic reagent (*D. melanogaster*) | '13XLexAop2-IVS-Syn-21-Chrimson::tdTomato (attP2)' | DOI: 10.1038/nmeth.2836 | | Barret Pfeiffer, Janelia Farm, HHMI |
| Genetic reagent (*D. melanogaster*) | 'P{y[+t7.7] w[+mC]=BDP-GAL4}attP2' | DOI: 10.7554/eLife.08758.027 | | Andrew Seeds, Steffi Hampel, UNIVERSITY OF PUERTO RICO |
| Genetic reagent (*D. melanogaster*) | 'P{w[+mW.hs]=GawB} VGlut[OK371]' | Bloomington *Drosophila* Stock Center | RRID:BDSC_26160 | |
| Antibody | nc82 (mouse monoclonal) | Developmental Studies Hybridoma Bank, | RRID:AB_2314866 | 1:50 |
| Chemical compound, drug | MLA | Tocris | TOCRIS_1029 | '1 µM methyllycaconitine citrate' |
| Software, algorithm | MATLAB | Mathworks | RRID:SCR_001622 | |
| Software, algorithm | DeepLabCut | DOI: 10.1038/s41593-018-0209-y | | Mathis Lab, Rowland Institute, Harvard University https://github.com/AlexEMG/DeepLabCut |

*Continued on next page*

*Continued*

| Reagent type (species) or resource | Designation | Source or reference | Identifiers | Additional information |
|---|---|---|---|---|
| Software, algorithm | FIJI | PMID:22743772 | RRID:SCR_002285 | |
| Software, algorithm | Fictrac | DOI: 10.1016/j.jneumeth.2014.01.010 | | |
| Other | Force probe fiber | Proform CS2.5 AS | This paper | "PBT fiber from a synthetic paint brush, cut to have a spring constant = 0.22 N/m. Find at any Proform retailer, e.g. Amazon.com' |
| Other | Green CST DPSS laser, | Besram Technology, Inc | | '532 nm' |
| Other | Phalloidin | Thermo FisherScientific | FISHER: A22284 | '633 nm, 1 unit per leg' |
| Other | streptavidin-Alexa Fluor | Thermo FisherScientific | FISHER: S11226 | '568 nm' |

## Fly husbandry

*Drosophila melanogaster* were raised on cornmeal agar food on a 14 hr dark/10 hr light cycle at 25° C and 70% relative humidity. We used female flies, 1–4 days post eclosion, for all experiments except tethered behavior experiments. Both male and female dark-reared flies, between 2–10 days post-eclosion, were used for tethered walking behavior experiments. For experiments involving optogenetic reagents (Chrimson variants and gtACR1), adult flies were placed on cornmeal agar with all-trans-retinal (100 µL of 35 mM ATR in 95% EtOH, Santa Cruz Biotechnology) for 24 hr prior to the experiment. Vials were wrapped in foil to reduce optogenetic activation during development.

## Electrophysiology and calcium imaging preparation

Flies were positioned in a custom steel holder as described in *Tuthill and Wilson, 2016b*, with modifications to allow us to image movement of the fly leg. Each fly was anesthetized on ice for two minutes, so that she could be positioned ventral side up with her head and thorax fixed in place with UV-cured glue. The front legs were glued to the horizontal top of the holder, the coxa aligned with the thorax, and the femur positioned at a right angle to the coxa and body axis. In this configuration, the fly could freely wave her tibia in an arc at an angle of ~50–65° to the top surface of the holder. The holder was placed in the imaging plane of a Sutter SOM moveable objective microscope. All recordings were performed in extracellular fly saline (recipe below) at room temperature.

We used a water immersion 40X objective (Nikon) for patch-clamping and a 5X air objective (Nikon) to view the fly's right front femur and tibia through the saline during spontaneous movements of the leg. Videos of the preparation were acquired at 170 Hz through the 5X objective with a Basler acA1300-200um machine vision camera. Custom acquisition code written in MATLAB (*Azevedo, 2020a*; copy archived at https://github.com/elifesciences-publications/FlySound) controlled generation and acquisition of digital and analog signals through a DAQ (National Instruments). Input signals were digitized at 50 kHz.

## Force measurement and spring constant calibration

To measure forces produced by leg movements, we imaged the position of a flexible 'force probe' as the fly pulled against it. The force probe was a PBT filament fiber from a synthetic paint brush (Proform CS2.5 AS), threaded through the end of a glass micropipette (1.5 mm OD, 1.1 mm ID, WPI). To create a force probe, UV-cured glue (KOA 300, KEMXERT) was sucked up into the micropipette, the fiber was threaded into the glue, leaving 1–2 cm protruding out from the tip of the glass, and the glue was cured. The micropipette allowed us to mount the force probe in a custom holder and to couple it to a micromanipulator (Sutter MP-285). Videos of the force probe were acquired at 170 Hz through the 5X objective with a Basler acA1300-200um machine vision camera. One pixel

equaled 1.03 µm². We wrote custom machine vision code that detects the position of the force probe in each frame of the video by 1) allowing the user to draw a line along the probe in the video, 2) rotate the image of the probe perpendicular to the line, 3) average down the rows of the rotated image to get a single intensity profile, with a peak at the probe's location, and then 4) find the center of mass of the intensity peak,±FWHM above baseline. A similar technique employing a probe to measure force has been used in *Drosophila* in previous studies (*Elliott and Sparrow, 2012*).

At steady state, the position of the force probe was related to the force through a spring constant, k, F = -kx (*Figure 1—figure supplement 2*). We measured the spring constant by positioning the force probe over an analytical balance. A glass coverslip was oriented vertically on edge in a piece of wax on the balance, and the tip of the force probe was positioned at the top edge of the coverslip. We then moved the micromanipulator to different positions. The 'mass' of the force probe was multiplied by gravity to give the force at that position. We then fit a linear relationship between force and position to measure the spring constant (*Figure 1—figure supplement 2A*). The force probe we used for experiments in this study had a spring constant of k = 0.2234 µN/µm and protruded approximately 1.5 cm past the end of the micropipette.

The force probe was not only a spring. It also had mass and was placed in saline, so inertia ($m$) and drag ($c$) affected its dynamics: $F = m\,d^2x/dt^2 + c\,dx/dt + kx$. To measure these properties, we 'flicked' the force probe by moving it to different positions with a glass micropipette, abruptly letting go, and allowing the probe to relax back to rest (*Figure 1—figure supplement 2B*). We imaged the position of the probe at 1.2 kHz with a restricted region of interest, and extracted dynamical parameters m = 0.1702 mg and c = 0.1377 kg/s. While not zero, the effective mass and drag were negligible (*Figure 1—figure supplement 2C*). The probe was slightly under-damped with a relaxation time constant of 2.5 ms and oscillation period of 5.8 ms, such that when imaging spontaneous movements at 170 Hz, the probe would effectively come to rest within one frame. The relaxation time constant was much faster than the fly's spontaneous movements, even when the fly was attempting to let go of the force probe.

In *Figure 4D–F*, we calculate force by including drag and inertia, but in other figures we report leg displacement and the approximation of force, assuming that drag and inertia are negligible. We easily captured the lateral movement of the force probe across the frame but avoided estimating the vertical movement as the fly pulled the probe closer to its leg. As a result, the displacement (and thus the force) measured by the probe in *Figures 1*, *5* may be slightly underestimated.

## Mechanical stimulation of the leg

To move the leg and passively stimulate proprioceptive feedback, we mounted the force probe perpendicularly on a piezoelectric actuator with a 60 µm travel range (Physik Instrumente). The axial position of the probe was controlled by an amplifier (Physik), with voltage commands generated in MATLAB and delivered through the DAQ board (National Instruments). The output of the actuator's strain gauge was used to control the position of the actuator through closed-loop feedback. The strain gauge sensor output was sampled at 50 kHz. The probe tip was positioned near the end of the tibia, giving a lever arm of 417 ± 7 (s.d.) µm across flies (n = 8). We then moved the tibia through its range of motion until it was approximately at 90° to the femur. To measure the effect of leg angle on the amplitude of sensory feedback, we then moved the probe to a range of axial positions (−150 µm = −21°, −75 µm = −10°, 75 µm = 10° and 150 µm = 21°, negative direction is extension) and repeated the stimuli (*Figure 6—figure supplement 1*).

We delivered ramp stimuli that moved the leg 60 µm (8°) with varying speeds, in both flexion (+) and extension (-) directions. We measured the actual speed of step stimuli by finding the maximum derivative of the strain gauge signal during the step onset. The range of angular velocities produced by the probe span the range shown to activate position- and velocity-sensitive femoral chordotonal neurons in the femur (*Mamiya et al., 2018*). Though the force probe was flexible, when we imaged the displacement of the force probe we saw that the probe tip matched the strain gauge feedback (errors < 5%), suggesting that passive or muscle forces did not impact these small stimuli.

To generate larger, faster movements than we could deliver with the probe, we whacked the leg by flicking the probe, similar to how we calibrated the probe dynamics (*Figure 6—figure supplement 1*).

## Leg tracking

In trials where the fly was free to wave its leg rather than pull on the force probe, we tracked the leg using DeepLabCut (*Mathis et al., 2018*). For a training set, we labeled the tibia position for ~45 frames for three different videos from each fly using custom labeling code (*Azevedo, 2020b*; copy archived at https://github.com/elifesciences-publications/LabelSelectedFramesForDLC) . We labeled six points on the stationary femur, six points on the tibia (*Figure 1—figure supplement 1B*), as well as prominent bright objects that would otherwise often be falsely identified as part of the legs, such as the EMG electrode, the force probe, and several specular creases in the steel holder. The resnet50 network used in DeepLabCut served as the starting point for training, but as we added more flies to the training set, we initiated further training from the previously trained network. We found that the networks failed to generalize across flies but that ~150 labeled frames were sufficient to ensure >99% accuracy on other frames for that fly (*Figure 1—figure supplement 1B*).

In post-processing, we measured the distribution of pairwise nearest neighbor distances between the six detected tibia points and assumed that outliers indicated that a point was poorly identified. If only a single point was misidentified (~0.7%), we filled in the point with random draws from the nearest-neighbor pairwise distance distributions. The network misidentified more than one point 0.2% of the time, typically when the fly moved its leg particularly quickly, causing the image to blur. We excluded such frames. We median-filtered the x, y coordinates across video frames, and found the centroid of the six points, approximately the middle of the tibia. The centroid points traced out an ellipse that was the projection of the circular arc of the leg in the plane of the camera. Fitting an ellipse to the centroids allowed us to calculate the azimuthal angle of the leg arc (~50–65°) and the real angle between the stationary femur and the moving leg. We then used the real angle of the leg to detect when the leg was extended (>120°) or flexed (<30°) (*Figure 1—figure supplement 1E*).

## Calcium imaging

We imaged the calcium influx into muscles with GCaMP6f (*Figure 1*, *Figure 1—figure supplement 1A*), driven by MHC-LexA expression in muscles. Epifluorescent 488 nm illumination excited GCaMP6f fluorescence. A long-pass dichroic (560 nm, Semrock) passed IR wavelengths to the leg imaging camera and reflected GCaMP6f emission to a second Basler camera imaging at 50–60 Hz. The imaging window of the GCaMP camera was restricted to the femur. Video frames were registered (*Guizar-Sicairos et al., 2008*) to remove vibrations due to movements of the saline meniscus (*Video 1*).

In the dark, the fly tended not to move its leg. However, the fly began to struggle and move its leg as soon as the blue epifluorescent light turned on to excite the calcium sensor. We imaged spontaneous movements under two conditions. 1) The fly could pull on the force probe with its tibia or 2) the fly waved its leg around spontaneously without the force probe. In the second case, a glass hook was placed near the femur as a barrier to prevent the fly from completely flexing the tibia and obscuring the calcium signals in the femur.

## Clustering of calcium signals

We used the k-means algorithm in MATLAB to segment calcium signals into clusters. We used trials in which the fly waved its leg with no force probe. We drew an ROI around the femur, excluding only points near the femur tibia joint where intensity changes were dominated by the tibia obscuring the femur. We used the correlation of pixel intensity as the distance metric and varied the number of clusters, k = 3–8. Once pixels were assigned to a cluster, we applied a Gaussian kernel to the pixel cluster assignments and excluded pixels which fell below 0.75, indicating that less than ¾ of the surrounding pixels were of the same cluster. This resulted in clearly defined clusters with separation between them but no major gaps, indicating that similar clusters were also grouped anatomically (*Figure 1—figure supplement 1C*). To confirm that the clustering indicated changes in calcium influx rather than muscle movement, we also ran the same clustering routines on flies expressing GFP in the muscles (*Figure 1—figure supplement 4*). In this case, clusters were dispersed, fluctuated very little in intensity and bore no similarity to musculature.

We found that six clusters produced broadly similar clusters across flies. We numbered the six clusters as follows: Cluster one was large and distal/ventral; Cluster two was proximal and ventral; Cluster three ran down the center of the leg, neighboring Cluster 2; Clusters 4, 5, six were assigned

from proximal to distal. With fewer than five clusters, the proximal cluster tended to be much larger, incorporating much of the region labeled as cluster 3 (*Figure 1—figure supplement 1F*). With more than six clusters, the smallest and least modulated clusters tended to divide, not giving us any further information. Six clusters potentially allowed for pixels that were most correlated with extension of the leg to cluster together, but we did not see large increases in fluorescence with extension.

We used the same clusters to measure fluorescence changes in trials where the fly pulled on the force probe. When the leg was flexed, the force probe could obscure the femur (e.g. *Figure 1—figure supplement 4* and *Figure 1—figure supplement 2*), so we included only proximal portions of the clusters. If the force probe still happened to obscure >40% of the pixels in a cluster, that cluster intensity for that frame was excluded from the analysis.

The time constant of the calcium indicator was slower than the fly's movements, such that fluorescence built up over subsequent contractions. Thus, pixel intensity (ΔF/F) was not directly related to contraction of the muscle. We took positive increases in cluster intensity to indicate muscle activation, i.e. neural activity. We applied a Sovitzky-Golay filter to interpolate cluster calcium signals (50 Hz) to the leg movements (170 Hz), which also computed the time-derivative of the local spline for each cluster (sgolay_t function in MATLAB, by Tiago Ramos, N = 7, F = 9). GCaMP6f decay was slow relative to leg movements (*Figure 1E* and *Figure 1—figure supplement 1E*), so negative derivatives reflected noise in the cluster fluorescence. We used this estimate of the noise (two standard deviations) to threshold the positive cluster derivatives, and thus find cluster 'activations'.

Surprisingly, we did not identify any clusters that increased their calcium activity during tibia extension. Flies occasionally held their legs extended (*Video 1*), at which point we expected to see some clusters increase fluorescence (*Figure 1—figure supplement 1D–E*). On average, the leg musculature was dim during these periods *Figure 1—figure supplement 1E*), whereas the fluorescence of superficial flexors muscle fibers could increase more than six-fold during flexion events. Diffuse emission from the bright and slowly fading flexors may have obscured small increases in extensor fluorescence. We still found it curious that contractions of extensors did not produce brighter events. We speculate that calcium influx and contractile forces may be larger in flexor muscles than extensors because flies use flexion of the forelimb tibia to support their weight, to hold onto the substrate, and to pull their body during walking, whereas extensors generally lift up unloaded limbs when swinging them forward.

## Whole-cell patch clamp electrophysiology

To perform whole-cell patch clamp recordings, we first covered the fly in a drop of extracellular saline and dissected a window in the ventral cuticle of the thorax to expose the VNC. The perineural sheath surrounding the VNC was ruptured manually with forceps, near the midline, anterior to the T1 neuromeres. We first used a large bore cleaning pipette (~7–10 μm opening) to remove debris and gently blow cell bodies apart, clearing a path from the ruptured hole in the sheath to the targeted motor neuron soma. The recording chamber was then transferred to the microscope and perfused with saline at a rate of 2–3 mL/min.

The extracellular saline solution was composed of (in mM) 103 NaCl, 3 KCl, 5 TES, eight trehalose, 10 glucose, 26 $NaHCO_3$, 1 $NaH_2PO_4$, 4 $MgCl_2$, 1.5 $CaCl_2$. Saline pH was adjusted to 7.2 and osmolality was adjusted to 270–275 mOsm. Saline was bubbled with 95% $O_2$/5% $CO_2$.

Whole-cell patch pipettes were pulled with a P-97 linear puller (Sutter Instruments) from borosilicate glass (OD 1.5 mm, ID 0.86 mm) to have approximately 5 MOhm resistance. Pipettes were then pressure-polished (*Goodman and Lockery, 2000*) using a microforge equipped with a 100X inverted objective (ALA Scientific Instruments). Polished pipettes had resistances of approximately 12 MOhms. The polished surface allowed for high seal resistances (>50 GΩ) to limit the impact of seal conductance on $V_{rest}$ (<1 mV). We used a Multiclamp 700A amplifier (Molecular Devices) for all recordings. The bridge resistance was balanced before sealing onto a soma. The pipette capacitance was compensated after the seal was made.

The internal solution for whole-cell recordings was composed of (in mM) 140 KOH, 140 aspartic acid, 10 HEPES, 2 mM EGTA, 1 KCl, 4 MgATP, 0.5 Na3GTP, 13 neurobiotin, with pH adjusted using KOH to 7.2 and osmolality adjusted to 268 mOsm. The neurobiotin was a generous gift from Rachel Wilson at Harvard Medical School.

The liquid junction potential for the whole cell recordings was −13 mV (*Gouwens and Wilson, 2009*). We corrected the membrane voltages reported in the paper by post hoc subtraction of the junction potential.

We applied 1 µM methyllycaconitine citrate (MLA, Tocris) to block cholinergic transmission in the VNC. To encourage spontaneous self-driven movements, we sometimes bath applied 0.5 mM caffeine (Sigma-Aldrich) during whole cell recordings, which prolonged periods of struggling in response to the epifluorescent light. *Figure 5* includes trials recorded in caffeine: 45 of the 90 trials from one fast cell, and 50 of the 100 trials from a second fast cell. Caffeine's effects have been reported to be due to dopamine receptor agonism (*Nall et al., 2016*). Caffeine application did not increase activity on the leg electrodes during periods of rest.

## Identifying Gal4 lines that label leg motor neurons

We screened the Janelia FlyLight collection (*Jenett et al., 2012*) to find Gal4 lines that sparsely label leg motor neurons. We obtained flies from the Bloomington *Drosophila* Stock Center (BDSC), and imaged leg expression of GFP to characterize muscle innervation patterns.

*Soler et al., 2004* described the leg musculature in *Drosophila*, which also serves as the basis for the leg motor neuron nomenclature (*Baek and Mann, 2009*; *Brierley et al., 2012*). For clarity, however, we refer to muscles of the femur as flexors or extensors, rather than as depressors or levators, which refer to the natural stance of the insect. Soler et al. in turn based their nomenclature on *Miller, 1950*. There appears to be a discrepancy between the two: the muscle named the tibia reductor muscle by Soler et al. is described as one of two depressor muscles by Miller, muscles 40 and 41. Miller applied the nomenclature of *Snodgrass, 1935* to *Drosophila* leg muscles. Muscles that control the tibia are located in the femur. Snodgrass described a reductor of the femur that was actually located in the trochanter, perhaps leading to this misreading. The trochanter-femur joint in most insects has limited mobility. Presumably, Snodgrass named it a reductor muscle because depressor or levator would be inaccurate. We agree with the characterization of Miller and Soler that muscles 40 and 41 are two distinct muscles as they attach via distinct tendons, as seen in X-ray images of the leg musculature (*Pacureanu et al., 2019*). Because the function of muscle 41 is to flex the tibia, we refer to it here as a tibia flexor.

## Extracellular recordings from leg motor neurons

We recorded electrical activity in the leg by inserting finely pulled glass electrodes (OD 1.5 mm, ID 0.86 mm) into the cuticle of the femur, taking care to avoid impaling the muscle. The electrodes were filled with extracellular saline. Extracellular currents were recorded in voltage clamp to improve the signal-to-noise. We confirmed that injecting currents of similar size did not move the leg nor produce additional electrical activity. The currents we recorded likely reflect spikes from the motor neuron axons, commensurate with the short latencies between somatic spikes and the events on the leg electrode (*Figure 4*). If we observed extracellular leg spikes that were time locked with whole-cell-recorded somatic spikes, we then never observed unmatched somatic spikes. If we were observing muscle action potentials or muscle EPSPs, we would have expected some failures. For simplicity we refer to activity recorded in the leg as the electromyogram (EMG), consistent with the use of the term in other organisms.

The content of EMG signals recorded in the leg depended on the placement of the electrode. To improve our chances of recording spikes from specific neurons, we used femur bristles (macrochaetes) as landmarks to place the electrode near the branched axon terminals. These include four large distal macrochaetes ('bristles 0–3') and one very proximal bristle which we call here the 'terminal bristle'. Fast motor neuron spikes were most likely found by impaling the leg between bristle 2 and 3, while large intermediate spikes were often found near the terminal bristle. Even still, the polarity, amplitude, and shape of events from identified neurons could vary substantially. When the electrode was placed near the third macrochaete, we tended to record very large units of >200 pA from the fast motor neuron. Fast neuron units were by far the largest amplitude events in the femur. When the electrode was placed in the proximal part of the femur, near the terminal bristle, the spikes from the fast neuron tended to be smaller but still identifiable. We could not unambiguously detect EMG units associated with Flexor 3 in *Figure 1*. We ran our spike detection routines (see

below) on EMG records only in cases where they could be clearly identified or when EMG spikes aligned with the somatic spikes.

## Optogenetic activation of leg motor neurons during electrophysiology

To measure force production as a function of motor neuron activity, we drove the expression of CsChrimson (*Klapoetke et al., 2014*) with 81A07-Gal4, and the expression of Chrimson88 (*Strother et al., 2017*) with 22A08-Gal4. CsChrimson expression in 22A08-Gal4 prevented straightening of the wings and caused the front legs to be bent midway through each segment. We drove expression of Chrimson (*Klapoetke et al., 2014*) in sensory neurons with *iav*-LexA. We activated Chrimson by placing a fiber-coupled cannula (105 μm diameter, Thorlabs) next to the ventral window in the cuticle and illuminating the T1 neuropil with a 625 nm LED (Thorlabs). We used short flashes of 10 or 20 ms to activate neurons, increasing intensity by increasing the voltage supplied to the LED driver (Thorlabs). We measured the power output of the LED for each voltage we used.

## Spike detection from whole-cell recordings and EMG recordings

To detect spikes in current clamp recordings of membrane potential, we applied the following analysis steps to our electrophysiology traces (digitized at 50 kHz): 1) filter, 2) identify events with large peaks above a threshold, 3) compare the shape of the filtered events to a template (distance metric), 4) threshold events based both on the shape and on the amplitude of the unfiltered spike. The parameter space for each of these steps was explored in an interactive spike detection interface (*Azevedo, 2020c*; copy archived at https://github.com/elifesciences-publications/spikeDetection) .

1. We high-pass filtered the first derivative of each trace with a 3-pole Butterworth filter with a cutoff at 209 Hz, a low-pass filter with a 3-pole Butterworth filter with a 898 Hz cutoff. Empirically derived, this procedure resulted in a large positive peak associated with the rapid reversal of the membrane potential at the top of a spike.
2. Events were identified by threshold crossing. Thresholds were in the range of 2–10 $*10^{-5}$ mV/s. The threshold was set as low as possible to reject baseline noise. Changes in membrane voltage associated with current injection and large secondary peaks associated with the filtered spike waveform oscillations often passed above threshold.
3. Events that passed threshold were then compared with a template using a dynamic time warping procedure over a 251 sample window centered on the event. The template was calculated for each cell, and though the amplitude varied with spike amplitude, the template time course (shape) was generally similar across cells and genotypes.
4. We then calculated the amplitude of the voltage fluctuation in the raw record associated with each event. Thus, each detected peak in the filtered data had a shape metric value and a spike amplitude. Events with both a similar shape (i.e. time course) as the template and a large amplitudes were identified as spikes.

We inspected records by eye to reject occasional false positives, such as changes in membrane voltage caused by current injection.

The algorithm was generally effective once the parameters were tuned for each cell. However, two cases typically caused spikes to become very difficult to unambiguously identify in slow motor neurons. First, during large current injections and the resulting depolarization, the spikes became very small and difficult to identify (*Figure 4*). High frequency spikes were clear in the raw voltage, and the leg moved, so we do not believe the neurons entered depolarization block. In such cases we hand tuned the parameters and inspected the identified spikes by eye to estimate the spike rate. Second, prolonged self-generated flexion could also depolarize slow motor neurons to the point that spike detection was difficult, we speculate because of large synaptic conductances that decreased input resistance. By contrast, spike detection worked perfectly during high spike rates evoked by sensory feedback (*Figure 6*).

Since we were interested in measuring spike latencies and conduction velocities, we calculated the second derivative of the raw voltage trace, smoothed over five samples, for each identified spike. We found the peak of the second derivative, using that point as the onset/acceleration of the spike.

We used the same algorithm to detect spikes from EMG current records, with different thresholding parameters in Step four for each recording due to the variability in EMG event size.

## Immunohistochemistry

After a whole-cell motor neuron recording, we dissected the VNC but left the legs and head attached. We placed the tissue in 4% paraformaldehyde in phosphate-buffered saline (PBS) for 20 min, then separated and retained the VNC and the right leg with the filled motor neuron.

The VNC tissue was washed in PBST (PBS + Triton, 0.2% w/w), placed in blocking solution (PBST + 5% normal goat serum) for 20 min, and then placed for 24 hr in blocking solution containing a primary antibody for neuropil counterstain (1:50 mouse anti-Bruchpilot, Developmental Studies Hybridoma Bank, nc82 s) and a streptavidin-Alexa Fluor conjugate to label the neurobiotin-filled motor neuron (Invitrogen). We washed the tissue again in PBST, and then placed the VNC in blocking solution containing secondary antibodies for 24 hr (1:250 goat anti-mouse Alexa Fluor conjugate from Invitrogen; streptavidin-Alexa Fluor).

The leg was first incubated in blocking solution, like the VNC. Then it was placed in PBST containing 0.01% sodium azide (Thermo Fisher), 1 unit of phalloidin (Thermo Fisher) and the streptavidin-Alexa Fluor, and allowed to incubate for two weeks at 4°C, with occasional nutation.

Following staining, the tissue was mounted in Vectashield (Vector Labs) and imaged on a Zeiss 510 confocal microscope (Zeiss). Cells were traced in FIJI (*Rueden et al., 2017*; *Schindelin et al., 2012*), using the Simple Neurite Tracing plug-in (*Longair et al., 2011*). Images in *Figure 2* show the results of the filling function in the FIJI plug-in. In some cases, the neuropil counterstain (anti-Bruchpilot) was omitted and the native autofluorescence of the tissue (along with nonspecific binding of streptavidin and GFP fluorescence) was used as reference.

To quantify morphology (*Figure 3*), we measured the soma diameter, the width of the neurite entering the neuropil, and the width of the axon, as close to the exit of the neuropil as possible. We made two measurements for each image and location and averaged the values.

## Fly preparation for walking experiments

Fly wings were clipped under cold anesthesia (<4 mins) 24 hr before walking experiments. the fly's dorsal thorax was attached to a tungsten wire (0.1 mm diameter) with UV-curing glue (KOA 300, KEMXERT). Tethered flies were given only water for 2–5 hr prior to being placed on the treadmill. In some experiments, the tethered flies were then decapitated under cold anesthesia and allowed to recover for 5–20 min prior to the experiment.

Intact tethered flies were positioned on a hand-milled foam treadmill ball (density: 7.3 mg/mm$^3$, diameter: 9.46 mm) that was suspended on a stream of air (5 l/min) and that freely rotated under the fly's movement (*Figure 7D*). The ball and fly were illuminated by three IR lights. In experiments on headless flies, we removed the spherical treadmill, leaving the flies suspended in air. For all trials, the temperature in the chamber was maintained between 26–28°C with a relative humidity of 58–65%.

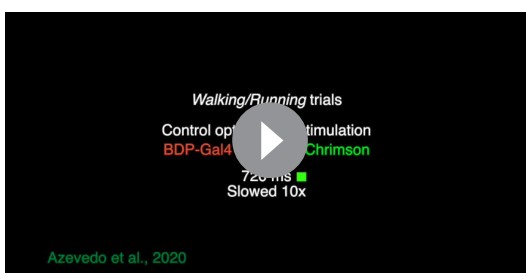

**Video 8.** Optogenetic stimulation of a control line (BDP-Gal4). The video shows the behavior of different control flies on different trials while 1) UAS-CsChrimson flies were walking; 2) UAS-CsChrimson flies were stationary; 3) UAS- gtACR1 flies were walking; 4) UAS-gtACR1 flies were stationary. Videos of behavior are slowed 10X.

https://elifesciences.org/articles/56754#video8

## Tethered behavior assay

We coaxed flies to walk on the ball by displaying visual stimuli on a semi-circular green LED display (*Götz and Wenking, 1973*; *Reiser and Dickinson, 2008*). To elicit forward walking, we displayed a single dark bar (width 30°) on a light background which oscillated across 48.75° about the center of the fly's visual field, at 2.7 Hz. During periods between trials, the LED panels displayed a fixed dark stripe (30°) on a bright background in front of the tethered fly.

To characterize the role of the motor neurons in behaving tethered flies, we optogenetically activated or silenced genetically targeted motor neurons. A green laser (532 nm, CST DPSS laser, Besram Technology, Inc), pulsed at 1200 Hz with a 66% duty cycle, passed through a converging lens and a pinhole (50 μm diameter) with a

resulting power of 87 mW/mm$^2$ at the target. It was aimed at the fly's left thoracic coxa-body wall joint, thus targeting the motor neuron axons and the T1 neuromere below the cuticle. Experiments using a driver line labeling all motor neurons (OK371-Gal4) indicated that optogenetic stimulation primarily effected neurons innervating the left prothoracic leg, though we cannot rule out effects on other VNC neurons (*Video 8*).

Each trial was four seconds long. We presented walking flies with the visual stimulus, the flies reached a steady running speed at ~1.5 s (*Figure 7E*), and the laser stimulus began at 2 s. We omitted the laser in some trials (0 ms), or the laser stimulus was either 90 ms or 720 ms in duration, interleaved in random order. Trials were separated by a 25 s period during which video data were written to disk and the LED panels displayed a fixed, stationary stripe.

## Quantification of fly behavior

We used Fictrac (*Moore et al., 2014*) to calculate fly walking trajectories (position, speed, and rotational velocity) from live video of the spherical treadmill's rotation (Point Grey Firefly camera, imaging at 30 Hz). Trajectories were then converted from pixels to mm using the spherical treadmill's diameter of 9.46 mm. Leg movements were captured from six simultaneously triggered cameras (Basler acA800-510µm, imaging at 300 Hz) that were spatial distributed around the fly's body. Digital and analog data signals were collected with a DAQ (PCIe-6321, National Instruments) sampling at 10 kHz and recorded with custom MATLAB scripts.

When headless flies were suspended above the ball, we manually tracked the position of the left femur and tibia during the 720 ms optogenetic stimulus period using manual annotation of video frames in FIJI (*Rueden et al., 2017*; *Schindelin et al., 2012*). We then calculated the femur-tibia joint angle from the position measurements in MATLAB.

When flies were in contact with the ball, we visually categorized the fly's behavior in the 200 ms preceding the optogenetic stimulus as 'Stationary', 'Walking/turning', 'Grooming' or 'Other'. Flies that took no steps for the duration of the categorization period were classified as Stationary. Flies that took at least four coordinated steps over the duration of the 200 ms period were classified as Walking/turning, irrespective of any distance traveled. Trials in which the fly switched behaviors, groomed or did not display clear markers for walking/turning during the categorization period were classified as Other/Grooming and excluded from analyses.

For Walking data, we calculated the average forward velocity over time, for each stimulus length, for each fly. We computed the percent change in walking speed for each fly by averaging walking speed during the stimulation (90 ms and 720 ms) and the subsequent 200 ms period, subtracting the average walking speed at stimulus onset, then dividing by the walking speed at stimulus onset. For Stationary trials, we calculated the percent of trials in which stationary flies reached steady state walking, *i.e.* sustained walking (>3 mm/s) in the half second following laser stimulation.

We noticed that the laser stimulus caused the empty Gal4 flies (BDP-Gal4) to decrease their walking speed slightly, likely because the flies could see the stimulus. This was particularly noticeable when we first used a fiber-coupled LED with a 105 µm diameter cannula. This prompted us to focus the laser on the fly's left leg in order to further reduce the spot size and minimize the behavioral artifact. Even so, the optogenetic stimulus increased the probability that stationary flies would start walking (*Figure 7H*). To control for this effect, we compared the change in speed in motor neuron lines for each stimulus duration, to the change in speed in the control empty Gal4 flies (see statistical analysis below). In no-light trials, walking initiation (*Figure 7H*, black and gray bars) and changes in speed (*Figure 7E*, black traces) did not vary across different genotypes, although baseline walking speed varied slightly across the different lines.

## Statistical analyses

For electrophysiology and calcium imaging results in *Figures 1–6*, no statistical tests were performed a priori to decide upon sample sizes, but sample sizes are consistent with conventions in the field. Unless otherwise noted, we used the non-parametric Mann-Whitney-Wilcoxon rank-sum test to compare two populations (e.g. *Figure 4*) and 2-way ANOVA with Tukey-Kramer corrections for multiple comparisons across three populations (e.g. *Figure 3*). To compare changes in fluorescence across multiple clusters and extension vs flexion (*Figure 1*), we used a 2-way ANOVA modeling an interaction between clusters and flexion vs. extension, with Tukey-Kramer corrections for multiple

comparisons. To compare cluster ΔF/F of multiple clusters (*Figure 1—figure supplement 1*), we used a 2-way ANOVA with Tukey-Kramer corrections for multiple comparisons. All statistical tests were performed with custom code in MATLAB.

For fly walking behavior in *Figure 7*, we used bootstrap simulations with 10,000 random draws to compare both the likelihood of stationary flies to start walking, as well as changes in walking speed (*Saravanan et al., 2019*). *Stationary* trials were assigned a binary value to indicate that the fly began walking (1) or not (0). For a given stimulus duration and optogenetic condition (Chrimson or gtACR), the binary values for the empty Gal4 control flies and a motor neuron line were combined and then drawn randomly with replacement in proportion to the number of trials for each genotype. As a metric, we measured the difference in the fraction of flies that began walking. The p-value was the fraction of instances in which the randomly drawn distribution produced a value of our metric more extreme then we saw in the data (two-tailed) (*Figure 7H*). For trials in which the fly was already walking at the onset of the laser stimulus (*Walking* trials, *Figure 7E*), we compared the relative change in speed following the stimulus for a given motor neuron line to the empty Gal4 line. We randomly assigned trials to each genotype and calculated the average speed change as above. We used the Benjamini-Hochberg procedure to calculate the false-discovery-rate for either activation or silencing.

## Table of genotypes

| | |
|---|---|
| *Figure 1A* | **W[*]; P{w[+mC]=Mhc-GAL4.K}2, P{y[+t7.7] w[+mC]=20XUAS-IVS-mCD8::GFP}attP2** |
| *Figure 1B* | w[1118]; P{JFRC7-20XUAS-IVS-mCD8::GFP} attp40/+; P{y[+t7.7] w[+mC]=GMR22A08-GAL4}attP2/+ |
| *Figure 1C–I* | w[1118]; MHC-LexA,w[13XLexAop2-IVS-GCaMP6f-p10]su(Hw)attP5/Berlin WT; +/Berlin WT; |
| *Figure 2* | w[1118]; P{JFRC7-20XUAS-IVS-mCD8::GFP} attp40/+; P{y[+t7.7] w[+mC]=GMR81A07-GAL4}attP2/+ w[1118]; P{JFRC7-20XUAS-IVS-mCD8::GFP} attp40/+; P{y[+t7.7] w[+mC]=GMR22A08-GAL4}attP2/+ w[1118]; P{JFRC7-20XUAS-IVS-mCD8::GFP} attp40/+; P{y[+t7.7] w[+mC]=GMR35 C09-GAL4}attP2/+ |
| *Figure 3* | w[1118]; P{JFRC7-20XUAS-IVS-mCD8::GFP} attp40/+; P{y[+t7.7] w[+mC]=GMR81A07-GAL4}attP2/+ w[1118]; P{JFRC7-20XUAS-IVS-mCD8::GFP} attp40/+; P{y[+t7.7] w[+mC]=GMR22A08-GAL4}attP2/+ w[1118]; P{JFRC7-20XUAS-IVS-mCD8::GFP} attp40/+; P{y[+t7.7] w[+mC]=GMR35 C09-GAL4}attP2/+ |
| *Figure 4* | w[1118]; P{JFRC7-20XUAS-IVS-mCD8::GFP} attp40/+; P{y[+t7.7] w[+mC]=GMR81A07-GAL4}attP2/P{y[+t7.7] w[+mC]=20 XUAS-IVS-CsChrimson.mVenus}attP2 w[1118], 10XUAS-syn21-Chrimson88-tDT3.1 (attP18); P{JFRC7-20XUAS-IVS-mCD8::GFP} attp40/+; P{y[+t7.7] w[+mC]=GMR22A08-GAL4}attP2/w[1118]; P{JFRC7-20XUAS-IVS-mCD8::GFP} attp40/+; P{y[+t7.7] w[+mC]=GMR35 C09-GAL4}attP2/+ |
| *Figure 5* | w[1118]; P{JFRC7-20XUAS-IVS-mCD8::GFP} attp40/+; P{y[+t7.7] w[+mC]=GMR81A07-GAL4}attP2/+ w[1118]; P{JFRC7-20XUAS-IVS-mCD8::GFP} attp40/+; P{y[+t7.7] w[+mC]=GMR22A08-GAL4}attP2/+ w[1118]; P{JFRC7-20XUAS-IVS-mCD8::GFP} attp40/+; P{y[+t7.7] w[+mC]=GMR35 C09-GAL4}attP2/+ |
| *Figure 6A–E* | w[1118]; P{JFRC7-20XUAS-IVS-mCD8::GFP} attp40/+; P{y[+t7.7] w[+mC]=GMR81A07-GAL4}attP2/+ w[1118]; P{JFRC7-20XUAS-IVS-mCD8::GFP} attp40/+; P{y[+t7.7] w[+mC]=GMR22A08-GAL4}attP2/+ w[1118]; P{JFRC7-20XUAS-IVS-mCD8::GFP} attp40/+; P{y[+t7.7] w[+mC]=GMR35 C09-GAL4}attP2/+ |
| *Figure 6G–H* | w[1118]; P{JFRC7-20XUAS-IVS-mCD8::GFP} attp40/iav-LexA; P{y[+t7.7] w[+mC]=GMR81A07-GAL4}attP2/13XLexAop2-IVS-Syn-21-Chrimson::tdTomato (attP2) w[1118]; P{JFRC7-20XUAS-IVS-mCD8::GFP} attp40/iav-LexA; P{y[+t7.7] w[+mC]=GMR22A08-GAL4}attP2/13XLexAop2-IVS-Syn-21-Chrimson::tdTomato (attP2) w[1118]; P{JFRC7-20XUAS-IVS-mCD8::GFP} attp40/iav-LexA; P{y[+t7.7] w[+mC]=GMR35 C09-GAL4}attP2/13XLexAop2-IVS-Syn-21-Chrimson::tdTomato (attP2) |
| *Figure 7* | w[1118]; +/+; P{y[+t7.7] w[+mC]=GMR81A07-GAL4}attP2/P{y[+t7.7] w[+mC]=20 XUAS-IVS-CsChrimson.mVenus}attP2 w[1118]; +/+; P{y[+t7.7] w[+mC]=GMR35 C09-GAL4}attP2/P{y[+t7.7] w[+mC]=20 XUAS-IVS-CsChrimson.mVenus}attP2 w[1118]; +/+; P{y[+t7.7] w[+mC]=GMR22A08-GAL4}attP2/P{y[+t7.7] w[+mC]=20 XUAS-IVS-CsChrimson.mVenus}attP2 w[1118]; +/+; P{y[+t7.7] w[+mC]=BDP-GAL4}attP2/P{y[+t7.7] w[+mC]=20 XUAS-IVS-CsChrimson.mVenus}attP2 w[1118]; +/+; P{y[+t7.7] w[+mC]=GMR81A07-GAL4}attP2/P{y[+t7.7] w[+mC]=20 XUAS-IVS- gtACR1}attP2 w[1118]; +/+; P{y[+t7.7] w[+mC]=GMR35 C09-GAL4}attP2/P{y[+t7.7] w[+mC]=20XUAS-IVS-gtACR1}attP2 w[1118]; +/+; P{y[+t7.7] w[+mC]=GMR22A08-GAL4}attP2/P{y[+t7.7] w[+mC]=20 XUAS-IVS- gtACR1}attP2 w[1118]; +/+; P{y[+t7.7] w[+mC]=BDP-GAL4}attP2/P{y[+t7.7] w[+mC]=20 XUAS-IVS- gtACR1}attP2 w[1118]; P{w[+mW.hs]=GawB}VGlut[OK371]/+; +/P{y[+t7.7] w[+mC]=20 XUAS-IVS- gtACR1}attP2 |

## Data and software availability

Data will be made available from the authors website. Acquisition code is available at https://github.com/tony-azevedo/FlySound. Analysis code is available at https://github.com/tony-azevedo/FlyAnalysis.

# Acknowledgements

We thank Jim Truman, Wei-Chung Allen Lee, Jasper Maniates-Selvin, Eiman Azim, Randy Powers, Marc Binder, and Brendan Lehnert, as well as members of the FlyLoops U19 team (Tom Clandinin, Michael Dickinson, Shaul Druckmann, Richard Murray, and Rachel Wilson) for helpful discussions, and members of the Tuthill laboratory for feedback on the manuscript. We thank Peter Detwiler, Fred Rieke, and Rachel Wong for generous sharing of equipment, Shellee Cunnington for preparation of solutions, Eric Martinson and Bryan Venema for technical assistance, and Michael Reiser and Michael Dickinson for sharing fly stocks. Stocks obtained from the Bloomington *Drosophila* Stock Center (NIH P40OD018537) were used in this study. We also acknowledge support from the NIH (S10 OD016240) to the Keck Imaging Center at UW, and the assistance of its manager, Nathaniel Peters. AWA was partially supported by NIH fellowship F32 DC013928. ED and PG were both supported by post-baccalaureate fellowships from the UW Institute for Neuroengineering (UWIN). LV was supported by NIH grant R01NS070644. This work was funded by a Searle Scholar Award, a Sloan Research Fellowship, a Pew Biomedical Scholar Award, and a McKnight Scholar Award to JCT, and NIH grant U19NS104655 to JCT and RS.

# Additional information

## Funding

| Funder | Grant reference number | Author |
|---|---|---|
| National Institutes of Health | U19NS104655 | Anthony W Azevedo<br>Evyn S Dickinson<br>Pralaksha Gurung<br>Lalanti Venkatasubramanian<br>Richard S Mann<br>John C Tuthill |
| Searle Foundation | Scholar Award | Anthony W Azevedo<br>Evyn S Dickinson<br>Pralaksha Gurung<br>John C Tuthill |
| McKnight Foundation | Scholar Award | Anthony W Azevedo<br>Evyn S Dickinson<br>Pralaksha Gurung<br>John C Tuthill |
| Pew Biomedical Trust | Scholar Award | Anthony W Azevedo<br>Evyn S Dickinson<br>Pralaksha Gurung<br>John C Tuthill |
| Alfred P. Sloan Foundation | Research Fellowship | Anthony W Azevedo<br>Evyn S Dickinson<br>Pralaksha Gurung<br>John C Tuthill |
| National Institute on Deafness and Other Communication Disorders | F32 DC013928 | Anthony W Azevedo |

The funders had no role in study design, data collection and interpretation, or the decision to submit the work for publication.

## Author contributions

Anthony W Azevedo, Conceptualization, Data curation, Formal analysis, Investigation, Methodology, Writing - original draft, Writing - review and editing; Evyn S Dickinson, Data curation, Formal

analysis, Investigation, Methodology; Pralaksha Gurung, Data curation, Investigation; Lalanti Venkatasubramanian, Richard S Mann, Resources; John C Tuthill, Conceptualization, Supervision, Funding acquisition, Investigation, Methodology, Writing - original draft, Project administration, Writing - review and editing

### Author ORCIDs
Anthony W Azevedo [ID] https://orcid.org/0000-0001-8318-9678
Evyn S Dickinson [ID] https://orcid.org/0000-0001-7518-9512
Lalanti Venkatasubramanian [ID] http://orcid.org/0000-0002-9280-8335
Richard S Mann [ID] http://orcid.org/0000-0002-4749-2765
John C Tuthill [ID] https://orcid.org/0000-0002-5689-5806

### Decision letter and Author response
Decision letter https://doi.org/10.7554/eLife.56754.sa1
Author response https://doi.org/10.7554/eLife.56754.sa2

## Additional files

### Supplementary files
• Transparent reporting form

### Data availability
All data is publicly available on Dryad https://doi.org/10.5061/dryad.76hdr7stb.

The following dataset was generated:

| Author(s) | Year | Dataset title | Dataset URL | Database and Identifier |
|---|---|---|---|---|
| Azevedo AW, Dickinson ES, Gurung P, Venkatasubramanian L, Mann R, Tuthill JC | 2020 | Data for: A size principle for recruitment of Drosophila leg motor neurons | http://dx.doi.org/10.5061/dryad.76hdr7stb | Dryad Digital Repository, 10.5061/dryad.76hdr7stb |

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
