## [Decision Letter]

**Acceptance summary:**

Azevedo et al. investigate how different motor neurons are coordinated during *Drosophila* leg movements and find that leg motor neurons are typically recruited following a "size principle." This work beautifully combines different techniques: Ca-imaging in muscles, electrophysiology in motor neurons, and genetic manipulations with behavioral read-outs including force generation that highlight *Drosophila* as a particularly advantageous system for determining the neural architectures and the mechanistic principles for the control of limbs.

**Decision letter after peer review:**

Thank you for submitting your article "A size principle for leg motor control in *Drosophila*" for consideration by *eLife*. Your article has been reviewed by three peer reviewers, and the evaluation has been overseen by Ronald Calabrese as the Senior Editor and Reviewing Editor. The following individual involved in review of your submission has agreed to reveal their identity: Chris Q Doe (Reviewer #1).

The reviewers have discussed the reviews with one another and the Reviewing Editor has drafted this decision to help you prepare a revised submission.

Summary:

In this paper, Azevedo et al. investigate how different motor neurons are coordinated during *Drosophila* leg movements. They found that leg motor neurons are typically recruited following a "size principle," where slower, smaller motor neurons are activated first, followed by larger more powerful motor neurons. As in vertebrates, the activation thresholds explain the order of recruitment. This work combines different techniques: Ca^2+^ imaging in muscles, electrophysiology in motor neurons, and genetic manipulations with behavioral read-outs including force generation. The limited number of motor neurons moving the fly leg make it an advantageous place to explore the control architecture and the principles are comparable to the more complicated vertebrate limbs.

Revisions:

All comments should be addressed; no further experiments are requested. We encourage the authors to revise expeditiously so we can publish quickly.

Reviewer #1:

1) I did not understand Figure 1C. What is the leg doing at the time of imaging? Is it flexing or extending during the 180 second imaging period?

2) Figure 2 does not show recruitment data for slow MNs, although it does for fast and intermediate MNs. I understand that Figure 5 covers this ground later, but it was distracting to have it missing in Figure 2. Perhaps adding foreshadowing text saying fast and intermediate are described here, but slow will be upcoming in Figure 5?

3) Please consider common labeling for Figure 1, 2, and 5 so that readers can track the same MNs/muscles through the paper. As it stands, the muscle groups are introduced in Figure 1 but they have different names in Figure 2. One possibility is to use the fast/intermediate/slow nomenclature even in Figure 1, but say "these muscle/MN groups are named fast/intermediate/slow for reasons that will be shown below."

4) The citation to the VNC TEM volume is distracting as no EM data are shown. If you have identified these MNs in the volume then it is probably worth showing them in a Supplemental Figure. Then readers can compare the neurobiotin morphology to the EM morphology. Without this, I don't see any value in citing that dataset.

5) The discussion is interesting but quite long, and a bit wandering in places. This is a very minor comment, but it might be helpful to copy edit the Discussion to reduce redundancy.

Reviewer #2:

The term "size principle" may not be immediately familiar to the broader scientific audience. I suggest expanding the title to "A size principle governs the recruitment order of *Drosophila* leg motor neurons."

The calcium imaging identifies different motor units, named Flexor 3, 2, and 1, where 1 is the last recruited, biggest, and most forceful. The numbering is a bit unfortunate unless they were aiming for a countdown. Do the activity clusters correspond exactly to specific muscles? The mapping between these regions, muscles, and motor neurons was tricky to follow. Also, Figure 1G,H lower panels are not explained in the caption.

There were some aspects of the Introduction and Discussion that I found challenging to follow. The authors could be encouraged to expand. Each muscle is actually innervated by several motor neurons. These data are from representative example motor neurons. The legs do need to activate these muscles in different order to perform different movements. There is some discussion that different pre-synaptic inputs and different proprioceptive feedback could explain the flexibility. The optogenetic activation and silencing experiments use light targeted to a single leg – what does this say about communication/coordination between legs? Flexion is a pulling movement – which is what the authors measure with their force probe – it would be nice to describe when in walking or other limb behaviors this pulling is used. To lift the leg for a step probably doesn't use all the force that can be generated. It would be nice to know what this looks like on the extension side, where the leg has to push off the weight of the body. A description of how to map between the experimental tests and the normal behavior would be helpful.

Figure 5E Explain the white dots more clearly in the caption: they indicate centroid of the 2D histograms of all frames both within and outside the 25 ms windows.

Figure 7 is confusing. In Figure 7A, the authors monitored femur-tibia joint angle during motor neurons activation in headless flies. Activation of slow neurons caused very little contractions followed by prolonged tibia extension. Authors mentioned that the extension is caused by resistance reflex for joint twist. But the same response was not observed in Figure 4C, where the current injection causes continuous tibia flexion. Is there an explanation for the difference? The activation and silencing of different motor neurons affect walking speed or probability of walking, with intermediate neurons contributing in a surprising way. Why does activation of intermediate neurons increase speed while slow and fast neurons slow it (7E)? Overall, Figure 7 could be improved to help connecting the walking behavior data to the organizational model suggested by the size principle.

Reviewer #3:

My comments are geared more on the conceptual and presentation side, outlined below in no particular order.

1) They divide the types of motor neurons into three classes. On one extreme is the large, fast, high force, low sensory feedback type; the other is small, slow low force, high proprioceptive feedback type; the third is the intermediate. How "true" is this subdivision? Is this a case like there are three distinct sizes of shirts---small, medium and large, or is it more a continuum and they just happen to have three motor neurons that represents three points along this continuum?

2) There is a strong emphasis on the size principle of muscle recruitment, with small neurons being recruited prior to large neurons, together with statements from their work that this principle can be violated (even the word dogma is invoked). I´m not an expert in this field, but I´m wondering whether this is the only conceptual framework that´s available for muscle recruitment. Are there competing theories? Maybe because I don´t know the history that I find it difficult to understand why the authors so heavily rely on this. Also, why the emphasis on this principle, when the assumption for the principle is likely not true in this case, i.e. the premotor inputs are likely different to the three motor neurons they are studying.

3) The description of the muscles and motor neurons are not so clear. Could they provide a figure where they outline all the muscles and all the motor neurons with their target muscle fibers, and then point out where the three motor neurons they are studying fits into this broad scheme (6 muscles, and at least 15 motor neurons; which of these are the three motor neurons discussed here) ? Also, they should try to be more consistent in how they refer to the muscles and movements, e.g. in Figure 1B, they say femur muscles, with flexors, etc. ; in 2A, they say tibia flexor/Flexor (I assume they mean femur muscles that move the tibia). This was a bit confusing at first. And when they say tibia flexor motor neuron, is this the motor neuron that innervates the femur muscle, whose action leads to tibia movement?

4) Figure 2 in general: there are separate letterings, but are these essentially one integrated figure? Meaning, e.g., does the left side of a,b,c,d refer to fast tibia flexor? If so, the presence of fly recording scheme in c is rather confusing; would be better put elsewhere so a consistent structure is maintained in the figure.

5) Figure 3. I realize they have the quantitation on the soma size, but it would be nice to actually have a picture for all three types as a visual comparison in addition to the numbers. Also, how do the size of the motor neurons correlate with size of the muscles/fibers? (see also point 3).

6) Figure 5F is a very helpful summary figure, but it could be made clearer. For example, the red marked Intermediate recruited lies just above the green Slow…; the legend can also be a bit more explanatory, e.g. start from low force and then to high, instead of just saying when fast spikes, slow and intermediate should already be firing.

7) Behavior description of walking phenotype: I think it would help if the description does not go from walking slowing down, and then saying it´s because the legs lock up. This seems trivial, but I was expecting that the walking rate would be slowing (of course if your leg is locked up, that´s going to affect your walking speed…). Is there a clear idea of the sequence of muscle activity that moves the thoracic leg during *Drosophila* adult walking or escaping? It might be helpful to provide a scheme of this if such knowledge exists.

8) They mention that there is a hierarchy of recruitment but now always. But do they see certain pairs that are always linked, and others that can be linked hierarchically but can be decoupled under different situations?

9) As an outsider and out of curiosity, how convincing is the argument that there are no GABA inhibitory neurons in *Drosophila*? There is a case of people saying the identity of the neurotransmitter used by the mushroom body kenyon cells is unknown, and after decades of saying it is a novel transmitter, one group found it to be cholinergic.

10) Finally, going back to point 2 above, the authors write in the discussion that a "key assumption of the size principle is that all motor neurons within a pool receive the same synaptic input…These data suggest that although the tibia flexor motor neuron pool may share some presynaptic inputs, they are not completely homogeneous." I found this rather striking because I never would even have thought to think it was homogeneous. With the known analysis of motor neuron inputs from connectomic work in other systems (e.g., *Drosophila* larval locomotion or feeding behaviors), it seems abundantly clear that different motor neurons innervating different muscles will not have homogeneous presynaptic inputs.

---

## [Author Response]

Revisions:All comments should be addressed; no further experiments are requested. We encourage the authors to revise expeditiously so we can publish quickly.

A theme across several comments was a request for clarity about the anatomy of the leg motor neuron system and how that anatomy relates to functional measures, particularly the calcium imaging in Figure 1. We agree with the reviewers that it is important to be clear about what is and is not known. This is particularly challenging because there are few prior studies of the *Drosophila* leg motor system.

Comments on the theme of “the neuroanatomy of the femur and how does Figure 1 relate to that neuroanatomy” include: reviewer 1’s comments 3, 4; reviewer 2’s comment 3; and reviewer 3’s comment 3.

Two key points of confusion were:

The labels “Slow”, “Intermediate”, and “Fast” describe motor neuron force production and recruitment order, but they do not indicate the number of neurons in each of these categories, nor whether there are strong boundaries between them.

The activity clusters we find through calcium imaging bear similarities to the morphology and recruitment order of specific motor neurons, but we are currently unable to definitively map the correspondence between muscle and motor neuron activity.

The concrete changes we have made to the Results section to better describe the neuromuscular anatomy include:

As we introduce the activity clusters in Figure 1, we state in the text that the numbering is arbitrary and generally moves dorsal to ventral. This was stated only in the Materials and methods before.

In the section where we introduce the motor neurons, we now define the proximal tibia flexor muscle as those fibers that connect to the tibia flexor tendon, whereas the distal tibia flexor muscle consists of fibers that flex the tibia but connect to other structures on the cuticle. We use this terminology to describe motor neuron innervation patterns and to interpret calcium imaging.

As we introduce the neurons in Figure 2, we rely on studies that used MARCM clone anatomy to estimate there is only 1 fast neuron; 2-5 intermediate neurons that innervate the proximal fibers of the proximal tibia flexor muscle; and 8-9 slow neurons that innervate the distal flexor muscle.

We make clear that “slow” ,”intermediate” and “fast” are both functional and anatomical designations, and they encapsulate the rank ordering of neuronal properties and recruitment order.

We clarify that we think at least two different intermediate neurons underlie signals in activity cluster Flexor 2 in Figure 1, while the single fast neuron governs the high force contractions in Flexor 1.

We clearly restate this model following measurements of force in Figure 4 and measurements from other flexor neurons.

Finally, we restate this model in the Discussion.

Our understanding of the architecture of the leg motor system is pieced together from three sources: MARCM clones of leg motor neurons, an x-ray tomography volume of the leg and VNC (Pacureanu et al., 2019. bioRxiv 653188), and a serialsection EM volume of the VNC (Maniates-Selvin et al., 2020).

We refer here and in our paper to the distal flexor muscle and proximal flexor muscle fibers. These are clearly distinct muscles in the sense that they attach to different tendons. All the proximal fibers attach to a single tendon (tibia flexor tendon), whereas the distal fibers connect to cuticular structures very close to where the flexor tendon attaches. The distal fibers originate on both sides of the leg and stretch to this point, forming a chevron shape to make room for the extensor muscle. To distinguish these more clearly, we refer in this response to the distal flexor muscle as muscle 41, and the proximal flexor muscle as muscle 40, as described in Miller, 1950.

Muscle 40 (the proximal flexor muscle) can in turn be divided into more distal muscle fibers and more proximal muscle fibers. The distal muscle fibers are controlled by the fast tibia flexor motor neurons, which we think corresponds to muscle activity cluster 1. The more proximal part of the muscle is controlled by intermediate tibia flexor motor neurons and corresponds to activity cluster 2 in Figure 1.

We are confident that the motor neuron we call “fast” is the only neuron that innervates cluster 1. It innervates long muscle fibers in the middle part of the femur, the distal part of muscle 40, that attach to the tibia flexor tendon. Extracellular recordings near those fibers reveal large spikes that always match one-to-one with spikes recorded from the cell body of the fast neuron (81A07-Gal4). This gives us confidence that we are not missing an additional fast motor neuron.

We believe the neuron that we call “intermediate” (22A08-Gal4), is one of two neurons that innervates the proximal muscle fibers of muscle 40, which connect to the same tendon as the “fast” muscle fibers connect to. We are certain there are at least two, the one labelled by 22A08-Gal4 that we extensively study, and one labelled by 81A06-Gal4 that is reported in Figure 4—figure supplement 1. 81A06-Gal4 labels ~6-8 motor neurons, so we did not use it extensively for recording. Of these two neurons, the one in 81A06-Gal4 leaves the nerve more proximally than the one in 22A08-Gal4, and it innervates the most proximal muscle fibers. The 22A08-Gal4 neuron innervates more distal fibers, but overlaps minimally with the fast neuron. In terms of force production, spikes in the 81A06-Gal4 flexor neuron produced small but visible twitches, similar to the 22A08-Gal4 intermediate neuron, but we are not certain about the relative amounts of force production.

We think that there are only two intermediate neurons that innervate the proximal fibers of the flexor muscle. Between the fast neuron and the two intermediate neurons, their axonal arbors appear to target most of the proximal muscle fibers. (Each muscle fiber is innervated by just one motor neuron.) We see this in our light microscopic images reported here and in the x-ray data. However, based on MARCM clone expression in the leg, Baek and Mann, 2009 found that six neurons – including the “fast” neuron, their LinH neuron – innervate the tibia flexor muscle, whereas Brierley and Williams, 2012, found only two, so it is possible that more may exist. In particular, such neurons likely innervate and control fibers that give rise to the calcium activity cluster, Flexor 3.

Finally, the motor neuron we define as slow innervates the distal tibia flexor muscle, muscle 41 (formerly known as the tibia abelled muscle). From the MARCM studies, there are 8-9 neurons that innervate this distal flexor muscle. We recorded from three other such neurons, shown in Figure 4—figure supplement 1. Two of them resemble the slow motor neuron in 35C09-Gal4 in that they are spontaneously spiking while the fly’s leg rests on the probe. They are quite similar to each other, both functionally and morphologically, so they might be the same neuron abelled by different Gal4 lines. The third rests below spike threshold but can be recruited by sensory stimuli. This third neuron was found accidentally, following a failed attempt to record from the fluorescently-labeled slow neuron 35C09-Gal4, so we will have to wait for additional tools to record from it systematically.

To summarize: 8-9 slow motor neurons innervate the distal tibia flexor muscle (muscle 41). 35C09-Gal4 is likely one of the weakest. 2-5 intermediate neurons innervate the proximal portion of the tibia flexor muscle that attaches to the tibia flexor tendon (muscle 40); 1 fast neuron innervates the rest of the tibia flexor muscle fibers (muscle 40).

How does motor neuron innervation then relate to the activity clusters we find in Figure 1? The activity cluster 1 reflects the distal portion of muscle 40, controlled by the fast neuron, whose activity can be seen in the EMG. Activity cluster 2 then reflects the proximal portion of muscle 40. However, we know this portion to be under the control of at least two different intermediate neurons, which we were not able to differentiate between. Finally, we cannot see a single cluster that looks like the distal flexor muscle 41. As we mention, the distal femur region is generally dim, the leg and cuticle are changing shape near the distal flexor, and the extensor muscle is moving and contracting within that region. Another explanation might be that the physiology of muscle 41 (low-force) is distinct from that of muscle 40 (high-force), including larger calcium concentrations during contractions.

We want to emphasize that the three motor neurons we focus on in this study cause flexion of the tibia. This is key to using the “size principle” as a framework. The size principle is the dominant, largely successful concept that describes the relationships between motor neurons that control the same direction of movement. It was formulated from studies of the gastrocnemius and soleus muscles in the cat. Those muscles both extend the foot, but they have separate origins and insertion points. The size principle thus can apply to multiple muscles, like our framework here, but not to antagonistic muscles, muscles in different segments of the leg, or muscle groups that have different effects on movements. It thus is not relevant for muscles under control of a single motor neuron, such as those in the *Drosophila* flight system.

The other muscles in the femur are the tibia extensor and the long tendon muscle. Both likely contribute to the calcium signals we record in Figure 1, but they are dominated by flexor activity and probably unresolvable without optical sectioning. What we know about the extensor and long tendon muscles comes from work in larger insects. The large body of work on the slow extensor tibiae (SETi) and the fast extensor tibiae (FETi) motor neurons help to establish our naming conventions, but our work shows why the terms are harder to apply to the more diverse flexor neurons. In this paper, we ignore the long tendon muscle entirely because its action is surely complex. These muscles are described in more detail in the references we cite, but we’d also like to stress that the work in *Drosophila* has been purely anatomical thus far, so we do not have functional measurements for extension or the role of the long tendon muscle.

Reviewer #1:1) I did not understand Figure 1C. What is the leg doing at the time of imaging? Is it flexing or extending during the 180 second imaging period?

To cluster pixels across coactivated regions, we imaged the fly leg during spontaneous, un-loaded movements; that is, this experiment did not involve a force probe. This meant that the force probe was not obscuring part of the leg, as depicted in Figure 1G, enabling us to visualize more of the leg.

We have changed the figure legend for Figure 1C to state: “K-means clustering of calcium signals (MHC-GAL4;UASGcamp6f) based on correlation of pixel intensities during 180 s of self-generated leg movements in an example fly (unloaded waving, see Materials and methods).” We also added a label to Figure 1C, and labelled Figure 1D to emphasize that the addition of the force probe means the leg movements are loaded.

2) Figure 2 does not show recruitment data for slow MNs, although it does for fast and intermediate MNs. I understand that Figure 5 covers this ground later, but it was distracting to have it missing in Figure 2. Perhaps adding foreshadowing text saying fast and intermediate are described here, but slow will be upcoming in Figure 5?

We believe the reviewer is referencing the fact that there is little EMG activity in the right-hand plot of Figure 2D, compared to EMG recordings during fast and intermediate neuron recordings. The intention behind showing the EMG in this case was to show that we could not record EMG signals associated with slow motor neuron spiking. This is important in order to make sense of the finding that the fly places some constant force on the force probe at rest, yet we see little activity in the EMG. This could have been more clear.

To clarify our point, we now specifically reference in the text and the figure legend the fact that EMG signals from the fast neuron are readily detectable. We had made this point only in the Materials and methods. We now also state in the Materials and methods and figure legend that we were unable to observe EMG events that correlated with the slow motor neuron, which was not made explicit before. We believe this is due to the axon diameter and the distal innervation of the axon terminals. The slow motor neuron connects to fibers near the femur-tibia joint, and we could not impale that region of the cuticle without interfering with movement of the leg.

3) Please consider common labeling for Figure 1, 2, and 5 so that readers can track the same MNs/muscles through the paper. As it stands, the muscle groups are introduced in Figure 1 but they have different names in Figure 2. One possibility is to use the fast/intermediate/slow nomenclature even in Figure 1, but say "these muscle/MN groups are named fast/intermediate/slow for reasons that will be shown below."

We are certain that the calcium activity cluster we call Flexor 1 represents the activity of the fast neuron. Flexor 2, on the other hand, likely represents the coordinated activity of at least two intermediate neurons. As we layout above, we have improved our explanations in the paper for how the neurons we describe relate to the activity clusters we find in Figure 1.

We feel the terminology introduced in Figure 1 is helpful for guiding the reader through that figure, so we try to do a better job indicating this is ultimately a loose relationship with the underlying neural activity. We appreciate the reviewer’s suggestion on how to clarify the text.

4) The citation to the VNC TEM volume is distracting as no EM data are shown. If you have identified these MNs in the volume then it is probably worth showing them in a Supplemental Figure. Then readers can compare the neurobiotin morphology to the EM morphology. Without this, I don't see any value in citing that dataset.

Per the reviewer’s advice, we have moved the citation of the EM work to the Discussion of the paper so as not to distract from the data we present here. The correspondence between EM connectomics and genetic strains is an ongoing effort, sure to be expanded upon in future projects.

5) The discussion is interesting but quite long, and a bit wandering in places. This is a very minor comment, but it might be helpful to copy edit the Discussion to reduce redundancy.

We thank Dr. Doe for his close reading, and we have attempted to streamline the Discussion. We eliminated discussion of spike timing, which will better accompany a more complete characterization of walking kinematics; we tightened up our discussion of violations of recruitment order and the implications for premotor circuitry; and we emphasized our understanding about how many fast, intermediate and slow neurons exist and their relationship to the musculature.

Reviewer #2:The term "size principle" may not be immediately familiar to the broader scientific audience. I suggest expanding the title to "A size principle governs the recruitment order of Drosophila leg motor neurons."

We appreciate the reviewer’s perspective and we have changed the title to "A size principle for recruitment of *Drosophila* leg motor neurons". We are hesitant to suggest that any of the properties that are encompassed in the term “size principle” are causal mechanisms, without the details of the synaptic input. The idea that neuron size and excitability alone can account for the recruitment order has been precisely the attractive but unresolvable hypothesis that has made the size principle a useful but unsatisfying concept.

The calcium imaging identifies different motor units, named Flexor 3, 2, and 1, where 1 is the last recruited, biggest, and most forceful. The numbering is a bit unfortunate unless they were aiming for a countdown. Do the activity clusters correspond exactly to specific muscles? The mapping between these regions, muscles, and motor neurons was tricky to follow. Also, Figure 1G,H lower panels are not explained in the caption.

We have added text to the figure legend to better explain what the lower panels in Figure 1G and H show. As for the numbering scheme, we take the reviewer’s point. This numbering scheme was somewhat arbitrary (generally from ventral to dorsal), yet it suggests an underlying structure. To address this and similar comments, our revisions to the text clarify the connection between the calcium imaging and the details of the underlying neural control, e.g that Flexor 2 is likely controlled by two neurons. We hope this underscores the numbering scheme, rather than the recruitment order we use the calcium imaging to demonstrate.

There were some aspects of the Introduction and Discussion that I found challenging to follow. The authors could be encouraged to expand. Each muscle is actually innervated by several motor neurons. These data are from representative example motor neurons. The legs do need to activate these muscles in different order to perform different movements. There is some discussion that different pre-synaptic inputs and different proprioceptive feedback could explain the flexibility. The optogenetic activation and silencing experiments use light targeted to a single leg – what does this say about communication/coordination between legs? Flexion is a pulling movement – which is what the authors measure with their force probe – it would be nice to describe when in walking or other limb behaviors this pulling is used. To lift the leg for a step probably doesn't use all the force that can be generated. It would be nice to know what this looks like on the extension side, where the leg has to push off the weight of the body. A description of how to map between the experimental tests and the normal behavior would be helpful.

Our study avoids detailed descriptions of walking kinematics, partly because most of the experiments involved a reduced preparation, and partly because several ongoing studies will take advantage of new 3D markerless tracking technology developed in our lab (anipose.org) to set a new benchmark for kinematics during walking. However, the reviewer is right that simple functional details could better set up the role of tibia flexion during typical behaviors, like those in Figure 7. To this end, we have distilled the following information into new statements in the Introduction and the Discussion, and we refer the reader to our videos of walking behavior.

The fly’s front legs stick out in front of them when they walk, and a typical stride involves placing the front tarsus on the ground/ball by flexing the tibia. It then *pulls* through by flexing the tibia. Then it flexes the femur to lift the leg (controlled by coxal neurons), and extends the tibia to reach forward before placing it down again. The rear legs, by contrast, extend the tibia to push the fly forward. This can be seen in our videos. The short example clip at anipose.org of tracked legs during walking is illustrative, but Bender, Simpson, and Ritzmann, 2010, show this in detail for cockroach legs. In our Discussion we cite studies that measure how the distribution of pulling and pushing forces change as ants and stick insects walks up inclines, for example, and flies will also likely shift how they walk as conditions change. As the reviewer notes, extension of the tibia (lifting of the leg) is unlikely to require the same force or the same graded control as flexion. Researchers have speculated this accounts for the greater number of flexor neurons (Sasaki and Burrows, “Innervation Pattern of a Pool of Nine Excitatory Motor Neurons in the Flexor Tibiae Muscle of a Locust Hind Leg.”).

In Figure 7, optogenetic activation of intermediate neurons increases walking speed, and silencing of intermediate neurons decreases the likelihood of starting to walk. The reviewer is right that these results will make more intuitive sense if we clearly state that flies flex their front tibia during each stride.

Figure 5E Explain the white dots more clearly in the caption: they indicate centroid of the 2D histograms of all frames both within and outside the 25 ms windows.

We have added this explanation to the Figure 5E legend.

Figure 7 is confusing. In Figure 7A, the authors monitored femur-tibia joint angle during motor neurons activation in headless flies. Activation of slow neurons caused very little contractions followed by prolonged tibia extension. Authors mentioned that the extension is caused by resistance reflex for joint twist. But the same response was not observed in Figure 4C, where the current injection causes continuous tibia flexion. Is there an explanation for the difference? The activation and silencing of different motor neurons affect walking speed or probability of walking, with intermediate neurons contributing in a surprising way. Why does activation of intermediate neurons increase speed while slow and fast neurons slow it (7E)? Overall, Figure 7 could be improved to help connecting the walking behavior data to the organizational model suggested by the size principle.

Connecting the results of Figure 1-6 with those of Figure 7 illustrates a general challenge facing the motor control field: we have characterized elements of the system under controlled conditions, and then we want to know how they work when sensorimotor feedback loops are free to influence their activity. Optogenetic activation during walking was the obvious experiment. The reviewer points out two of the more confounding findings, 1) why does slow motor activation strongly disrupt walking, and 2) why does intermediate neuron activation cause walking? Because the experiment is somewhat orthogonal to the organizational model, it results in a messy narrative structure. To improve this, we have changed the text connecting Figures 1-6 to Figure 7 to say that feedback loops are now in play and that the experiment should really be seen as testing the effects of activating neurons outside of their normal recruitment patterns and outside of the fly’s control.

In that sense, we feel like our findings in Figures 1-6 better explain the effect of activating and silencing intermediate neurons. We know the intermediate neurons produce forces on the scale of ~1/10th of the fly’s weight and are quite fast. So, if the fly does not intend to recruit them, their activation creates a sudden but not disruptive perturbation on the ball, which is dealt with by walking forward. This is also consistent with tibia flexion moving the fly forward, so that general information would be helpful to have, see response above.

We were more puzzled that optogenetic activation of slow neurons causes tibia extension in the walking behavior experiments. We placed the same flies in the force probe setup, though we did not simultaneously patch the slow neurons. For the most part, we observed the fly flexing against the force probe, as the fly did following current injection at the cell body in Figure 4. With higher light intensities, this pulling could be more forceful than we observed in whole cell recordings, suggesting that chrimson activation could be more effective at changing the spike rate than current injection, as we observed for the fast and slow neuron. We observed extension in some instances when the leg was unloaded (no force probe). We did occasionally observe extension with the loaded leg. Since the slow neuron innervates fibers on one side of the leg, we speculate that in the unloaded leg, lateralized activation of muscle fibers causes a twisting motion that the fly reacts to.

We think a careful examination of the differences between current injection and Chrimson activation will involve more biophysical modelling of motor neurons and is beyond the scope of the paper. Generally, this approach might address why current injection at the soma is so ineffective, particularly in fast and intermediate neurons. More importantly, however, biophysical modeling would allow us to understand the intrinsic conductances in the motor neurons and integration of inputs, a critical part of going beyond the size principle framework for peripheral motor control.

Reviewer #3:My comments are geared more on the conceptual and presentation side, outlined below in no particular order.1) They divide the types of motor neurons into three classes. On one extreme is the large, fast, high force, low sensory feedback type; the other is small, slow low force, high proprioceptive feedback type; the third is the intermediate. How "true" is this subdivision? Is this a case like there are three distinct sizes of shirts---small, medium and large, or is it more a continuum and they just happen to have three motor neurons that represents three points along this continuum?

The reviewer has vividly posed an important question. Above, we have tried to clarify what we know about the divisions of motor neurons, but we can repurpose the reviewer’s metaphor to restate. We think the slow motor neurons are like a box of assorted sizes of small t-shirt, some close fitting v-necks down to skimpy tube tops (like our slow motor neuron), whereas the intermediate neurons are a separate box of 2-5 long sleeve shirts. The fast neuron would then be like a unique XXL sweatshirt.

2) There is a strong emphasis on the size principle of muscle recruitment, with small neurons being recruited prior to large neurons, together with statements from their work that this principle can be violated (even the word dogma is invoked). I´m not an expert in this field, but I´m wondering whether this is the only conceptual framework that´s available for muscle recruitment. Are there competing theories? Maybe because I don´t know the history that I find it difficult to understand why the authors so heavily rely on this. Also, why the emphasis on this principle, when the assumption for the principle is likely not true in this case, i.e. the premotor inputs are likely different to the three motor neurons they are studying.

In our understanding of the literature, the size principle is not only a conceptual framework, it is a well-documented empirical description that links the electrical properties of motor neurons with the gradients of force production across motor units. It was an early success in understanding the physiology of motor neurons, generally, and in describing structure in a population of neurons. The corresponding gradients of excitability and force production also suggested exquisite developmental processes to establish these linked properties, and spurred on the search for similarly compact descriptions elsewhere in the brain. The linked gradients at the core of the concept have been found in vertebrates (cats, fish, humans, etc) and in each systematic investigation of motor pools in invertebrates. As a shorthand for an intricate set of properties, the size principle is thus the dominant concept for motor neurons and motor control in the periphery in animals with limbs. As noted above, the size principle only applies to neurons controlling the same movement, such as when muscle fibers all connect to the same tendon, but still, this helps to collapse a lot of possible arrangements into an orderly low dimensional arrangement. In short, there is no competing conceptual framework for this situation, and since we find that it applies to *Drosophila* as well, we use it.

However, as a successful and dominant concept in the field, the size principle is also ripe for oversimplification, namely that a gradient of excitability alone governs the recruitment order. This is a possibility, but the relevant information required to augment the theory is an understanding of the different inputs to neurons within the same pool. We agree with the reviewer that it seems unlikely all the neurons in a pool get similar input, even when a motor pool controls one direction of movement in a rigid limb, but it remains to be rigorously tested. This should be possible in flies once we identify the leg motor neurons in the EM volume and map their inputs, a non-trivial but doable project.

Work in the *Drosophila* larva has paved the way for this approach (Zarin et al., “A Multilayer Circuit Architecture for the Generation of Distinct Locomotor Behaviors in *Drosophila*.”). There, each motor neuron receives distinct inputs, together creating patterns of contraction that distort the larva’s soft body. The same types of gradients in neuron properties have not been found in larvae (Zwart et al., “Selective Inhibition Mediates the Sequential Recruitment of Motor Pools.”)

Detailed mapping of motor neuron inputs has not yet been possible in vertebrates, though. The scale of the problem (e.g. 600 neurons controlling extension of a cat’s foot) is a large and the task daunting. As a result, attempts to modify the concept of a recruitment order – specifically one governed simply by motor neuron properties – have been limited to finding phenomena that cannot be explained by this simple mechanism. We find it notable that we have also seen similar phenomena in the fly, e.g. instances when the recruitment order was reversed, increasing the likelihood that what we find will apply to organisms with larger nervous systems. We look forward to the day when we can conceptualize motor neuron recruitment as the result of structured inputs to a structured neuromuscular periphery.

3) The description of the muscles and motor neurons are not so clear. Could they provide a figure where they outline all the muscles and all the motor neurons with their target muscle fibers, and then point out where the three motor neurons they are studying fits into this broad scheme (6 muscles, and at least 15 motor neurons; which of these are the three motor neurons discussed here) ? Also, they should try to be more consistent in how they refer to the muscles and movements, e.g. in Figure 1B, they say femur muscles, with flexors, etc. ; in 2A, they say tibia flexor/Flexor (I assume they mean femur muscles that move the tibia). This was a bit confusing at first. And when they say tibia flexor motor neuron, is this the motor neuron that innervates the femur muscle, whose action leads to tibia movement?

The introduction to our rebuttal attempts to address the main concerns raised in this comment (see above). We are still working towards a complete and accurate understanding of the detailed neuroanatomy and biomechanics of the leg. We have changed the text as indicated above to clarify what we know. Our best information on biomechanics comes from light microscopy studies and is limited, but newer datasets such as the XRay volume we reference will help. Our understanding of leg neurons comes from two studies of MARCM clones, a technique that can give repeated examples but not a complete overview of the system.

We have used established terminology where appropriate, but we have made a few changes. Specifically, in the literature “tibia flexor” neurons are called “tibia depressor” neurons, since they tend to lower the tibia when the insect is standing. “Depressor” is a confusing term, in part because the fly is inverted in our experiments. The term “tibia flexor neuron” is thus a compromise that better explains what the neuron is doing. As the reviewer notes, this can cause confusion, even to us and we found several instances where we referred incorrectly to “femur flexors”. We have edited the text to clearly state the name of each neuron, its function, and its innervation of the femur.

4) Figure 2 in general: there are separate letterings, but are these essentially one integrated figure? Meaning, e.g., does the left side of a,b,c,d refer to fast tibia flexor? If so, the presence of fly recording scheme in c is rather confusing; would be better put elsewhere so a consistent structure is maintained in the figure.

We experimented with moving the schematic to Figure 3, but the recording configuration was important to have for interpreting panels C (biocytin fills) and D (whole-cell recordings). We have added a shaded box to the schematic inset, to make it more clear that the traced neurons should be compared.

5) Figure 3. I realize they have the quantitation on the soma size, but it would be nice to actually have a picture for all three types as a visual comparison in addition to the numbers. Also, how do the size of the motor neurons correlate with size of the muscles/fibers? (see also point 3).

We have added a supporting figure showing the images from which the traces in Figure 2C were created. How the size and properties of the muscle fibers relates to the number of fibers in a motor unit and the force generated will be a fascinating future direction. Light microscopy cannot sufficiently resolve the edges of each muscle fiber to be able to segment them, currently. EM and XRay approaches will improve this going forward. Muscle ATPase, kinases and components of metabolism are likely to differ across muscle fibers, as do sarcomere lengths, which we observe.

6) Figure 5F is a very helpful summary figure, but it could be made clearer. For example, the red marked Intermediate recruited lies just above the green Slow…; the legend can also be a bit more explanatory, e.g. start from low force and then to high, instead of just saying when fast spikes, slow and intermediate should already be firing.

We have taken the reviewer’s suggestion and changed the legend to walk through this recruitment order schematic starting from low force to high force.

7) Behavior description of walking phenotype: I think it would help if the description does not go from walking slowing down, and then saying it´s because the legs lock up. This seems trivial, but I was expecting that the walking rate would be slowing (of course if your leg is locked up, that´s going to affect your walking speed…). Is there a clear idea of the sequence of muscle activity that moves the thoracic leg during Drosophila adult walking or escaping? It might be helpful to provide a scheme of this if such knowledge exists.

Reviewer 2 raised a similar concern above, helping us to realize that we did not clearly state that the front tibia flexes during normal walking to pull the fly forward, while the tibia of the rear legs extends to push the fly forward. We have added this important information to the Introduction and Discussion. We have also taken the reviewer’s suggestion and clarified our interpretation of the effect of stimulating the fast motor neuron.

8) They mention that there is a hierarchy of recruitment but now always. But do they see certain pairs that are always linked, and others that can be linked hierarchically but can be decoupled under different situations?

In short, all three neurons we studied, together with the examples from Figure 4—figure supplement 1, do appear hierarchically linked, but we also see instances that suggest that order can be softened. Just to restate the result from our paired recordings in Figure 5—figure supplement 1C-D, whenever the fast neuron was spiking, we observed spikes in the intermediate neuron, suggesting this hierarchy was strict. In the vast majority of cases, we found that slow neuron spikes preceded intermediate neuron spikes. However, in a few cases, intermediate neuron spikes were not directly preceded by slow spikes (110/3082).

To expand on this result, in those cases where we observed intermediate spikes before slow spikes, the *slow neurons were spiking*, but were just slower to spike and with lower instantaneous rates. These instances occurred when the fly was rapidly moving its leg. If, as we lay out above, the slow neuron is a member of 8-9 neurons controlling the distal flexor muscle, then perhaps we would see a similar reordering of spikes if we could record from the intermediate neuron along with some of those other slow neurons, like those in Figure 4—figure supplement 1.

We draw attention in the text to similar notable findings in zebrafish when the fish are swimming very quickly. In those cases the slow neurons drop out completely. Both cases suggest that when the organism is trying to make some rapid, alternating movement, slowly contracting muscles might get in the way and it would be better to de-recruit them or not drive them as hard. A simple mechanism could be selective inhibition to slow motor neurons that accompanies motor commands to move quickly.

As a final note, we also observed that suddenly turning on the epifluorescent lamp could elicit a spike in the fast neuron in some rare cases. We could not replicate this event with any regularity, we think it would happen when the epifluorescent lamp had not been turned on for some time and the fly was startled. Time was of the essence during whole-cell recordings, so we did not test this. It would make sense for there to be a pathway for rapid recruitment of the fast neuron: flexion of the tibia would be required to anchor the fly to the substrate, if, say, a strong breeze threatened to carry the fly away. We mention it because it is possible that this startle response also recruits the fast neuron out of order. One motif to look for in EM connectomics work would be descending neurons from the brain that might preferentially synapse onto the fast neuron but not the intermediate neuron.

9) As an outsider and out of curiosity, how convincing is the argument that there are no GABA inhibitory neurons in Drosophila? There is a case of people saying the identity of the neurotransmitter used by the mushroom body kenyon cells is unknown, and after decades of saying it is a novel transmitter, one group found it to be cholinergic.

We are relatively convinced that *Drosophila* have no inhibitory motor neurons. In insects with inhibitory neurons, the neurons are GABAergic. The most obvious way we thought of to see inhibitory motor neurons was to image expression of GABAergic markers in the leg. When we did this in preliminary studies with the Gad1-Gal4 line we saw no expression in the leg, and we didn’t even take images (regretfully). So, while the Gad1-Gal4 line labels many GABAergic neurons in the VNC, none of them appear to innervate muscles in the leg.

More convincingly, work on the development of neurons in the VNC has not detected inhibitory MNs, specifically from the neuroblast (NB5-5) that gives rise to them in other insects (Schmid, Chiba, and Doe, “Clonal Analysis of *Drosophila* Embryonic Neuroblasts.”).

10) Finally, going back to point 2 above, the authors write in the Discussion that a "key assumption of the size principle is that all motor neurons within a pool receive the same synaptic input…These data suggest that although the tibia flexor motor neuron pool may share some presynaptic inputs, they are not completely homogeneous." I found this rather striking because I never would even have thought to think it was homogeneous. With the known analysis of motor neuron inputs from connectomic work in other systems (e.g., Drosophila larval locomotion or feeding behaviors), it seems abundantly clear that different motor neurons innervating different muscles will not have homogeneous presynaptic inputs.

We completely agree with the reviewer that the assumption of homogeneous inputs is quite unlikely. In particular, as the reviewer notes, neurons innervating different muscles are sure to have different inputs, as is the case in the larva, or the opposing resistance reflexes in flexor vs extensor motor neurons (Burrows, *The Neurobiology of an Insect Brain*).

The question that applies here is, do neurons that control the same movement – either through innervation of different muscles (slow vs. intermediate and fast), or through innervation of distinct muscle fibers of the same muscle (fast vs. intermediate) – do they receive input from the same set of premotor neurons? Believe it or not, this has been a persistent question, mainly because the precise input to such a motor pool has not been mapped in any system as it has in the larva (and perhaps *C. elegans*).

All three of the neurons we study here are depolarized by extension of the leg, and some portion of that response is likely to come through a common pathway from proprioceptors to motor neurons. What portion, and what the rest of the input does to coordinate recruitment, is precisely the question our study will allow us to ask in the future, with the tools we report here. Fortunately, our study recapitulates several of the (surprisingly sparse) findings in vertebrates that indicate that some sources of input are specific to distinct members of a motor pool, such that recruitment order can be modified. This suggests that addressing this question in flies will have broad impact.

The unanswered question of whether the input to a motor pool is homogeneous, as it is referred to in the vertebrate motor neuron literature, is really about whether identical information about motor state and commands is conveyed to all members of the motor pool (same direction of movement). Given the measured gradient of excitability in a motor pool, an attractive model has been that more neurons could be recruited as that hypothetically identical input gets larger. Based in part on this fundamental finding, theoretical studies of neural networks often include a gradient of spike thresholds and time constants, in networks of linear integrate and fire neurons for example (Gerstner et al., *Neuronal Dynamics*), which simply indicates the usefulness of this concept. We agree with the reviewer that this ignores what is likely to be an important part of the puzzle. Frankly, in the past, authors have often used the possibility of identical inputs, as we have, as a simplifying assumption just to indicate what we do not yet understand.

To illustrate an analogous assumption at work in larval motor control, the field assumes the circuit motifs in a given segment of the larva will repeat in neighboring segments. This logical simplification rests on detailed development work and permits modelling efforts to understand coordinated contraction within and across segments. Interestingly, recent work on larval motor control relies on a reconstruction of A1, which receives different input than other abdominal segments, requiring additional assumptions in order to model a repeating structure (Zarin et al., “A Multilayer Circuit Architecture for the Generation of Distinct Locomotor Behaviors in *Drosophila*.”). Likewise, the distinct proprioceptive input to fast, intermediate and slow neurons, or the reversal of recruitment order in select cases, both indicate that the functions of the motor neurons within a recruitment hierarchy can diverge and spurs us on to find mechanisms underlying these differences.